DOI: 10.1038/s41467-018-05398-9　　**OPEN**

# Mechanical mismatch-driven rippling in carbon-coated silicon sheets for stress-resilient battery anodes

Jaegeon Ryu [1], Tianwu Chen [2], Taesoo Bok[1], Gyujin Song[1], Jiyoung Ma[1], Chihyun Hwang[1], Langli Luo [3], Hyun-Kon Song[4], Jaephil Cho[1], Chongmin Wang [3], Sulin Zhang[2] & Soojin Park [1]

High-theoretical capacity and low working potential make silicon ideal anode for lithium ion batteries. However, the large volume change of silicon upon lithiation/delithiation poses a critical challenge for stable battery operations. Here, we introduce an unprecedented design, which takes advantage of large deformation and ensures the structural stability of the material by developing a two-dimensional silicon nanosheet coated with a thin carbon layer. During electrochemical cycling, this carbon coated silicon nanosheet exhibits unique deformation patterns, featuring accommodation of deformation in the thickness direction upon lithiation, while forming ripples upon delithiation, as demonstrated by in situ transmission electron microscopy observation and chemomechanical simulation. The ripple formation presents a unique mechanism for releasing the cycling induced stress, rendering the electrode much more stable and durable than the uncoated counterparts. This work demonstrates a general principle as how to take the advantage of the large deformation materials for designing high capacity electrode.

[1] Department of Energy Engineering, School of Energy and Chemical Engineering, Ulsan National Institute of Science and Technology (UNIST), 50 UNIST-gil, Ulsan 44919, Republic of Korea. [2] Department of Engineering Science and Mechanics, Pennsylvania State University, University Park, PA 16802, USA. [3] Environmental Molecular Sciences Laboratory, Pacific Northwest National Laboratory, 902 Battelle Boulevard, Richland, WA 99354, USA. [4] Department of Chemical Engineering, School of Energy and Chemical Engineering, Ulsan National Institute of Science and Technology (UNIST), 50 UNIST-gil, Ulsan 44919, Republic of Korea. These authors contributed equally: Jaegeon Ryu, Tianwu Chen, Taesoo Bok. Correspondence and requests for materials should be addressed to C.W. (email: chongmin.wang@pnnl.gov) or to S.Z. (email: suz10@psu.edu) or to S.P. (email: spark@unist.ac.kr)

Stress management of electrode materials poses a major challenge for a stable operation of lithium-ion batteries (LIBs)[1–3]. High degree of lithium (Li) intake in promising lithium alloying materials (i.e., Si, Ge, Sn, etc.) leads to dramatic volumetric increase and internal stress generation, and eventually crack nucleation and propagation[4–7]. This typical chemomechanical degradation mechanism has been extensively investigated for Si electrodes due to their anomalous swelling when delivering a high specific capacity of 3572 mA h g$^{-1}$ (Li$_{15}$Si$_4$)[8–11]. Based on previous studies, several material designs for effective mitigation of the stress-driven battery material degradation have been proposed: (i) Nano-engineering Si anodes below the critical fracture size (~150 nm), for which the large surface-to-volume ratio facilitates stress relaxation and improves the fracture resistance on the particle level[12–17]; (ii) Nanocompositing with inactive/active buffers and building up a secondary structure with an increased tap density[18–21]; (iii) Using elastic and robust binders for enhanced structural integrity during electrochemical cycling[22–24]; (iv) Using porous structures to accommodate excessive volume expansion during lithiation[25]. These approaches have achieved significant success in battery performances, particularly cycle stability and rate capability. Yet, with only a few exceptions, the aforementioned materials failed to attain a high initial Coulombic efficiency (ICE), as well as improved tap density while maintaining its stress-releasing action for robust battery operation.

From the fundamental understanding on nanostructured Si, two-dimensional (2D) Si facilitates large-area (in-plane) preparation, while its nanoscale thickness (out-of-plane) mitigates stress generation during lithiation (discharge). Similarly, Si thin film has long been considered as a suitable electrode configuration except for its low loading density, poor electric conductivity (~10$^{-4}$ S m$^{-1}$), and absence of binders[6,26]. Recently, several synthetic methods for 2D Si have been established (e.g., chemical vapor deposition (CVD)[27], templated coating method[28], molten salt-induced exfoliation[14], and exfoliation of lithiated Si[29]) and showed the promise of achieving high-energy-density LIBs. These studies call for fundamental understanding in stress evolution and chemomechanical degradation toward optimized performance of 2D Si.

Si nanostructures coated with various layers of different chemomechanical characteristics can suppress the volume change and prevent particle pulverization[30–34]. Amorphous carbon (C) layers prepared via the CVD method also provide a conductive path for fast ionic transport. Further, the carbon layer can act as an interfacial barrier that prevents direct penetration of liquid electrolyte into the Si anode. However, repeated expansion/contraction during battery cycling results in a sponge-like morphology in Si, which mixes with carbon layers in most nanostructures regardless of their size[35]. This type of C-coated Si nanomaterials usually pulverizes and follows the general chemomechanical failure mode without stress-releasing actions, except for the Si yolk–shell structures with pre-reserved void space[36]. To date, diverse designs of Si anodes have geared towards releasing the lithiation-induced stress but failed to retain cyclability and durability in real battery systems.

In this work, we report observations of electrochemically cycling induced rippling in 2D Si nanosheets conformally coated with amorphous carbon layers (2DSi@C) by in situ transmission electron microscopy (TEM) analysis. During the delithiation (charge) process, the large deswelling ratio of the two material components of Si/C and distinctive structural characteristics of 2D materials induce the peculiar rippling morphology. Our in situ experiments and mechanics analyses reveal that owing to the rippling morphology internal stress of 2DSi@C is significantly reduced during the subsequent cycles, presenting a novel

mechanism for improved mechanical durability. In contrast, bare 2DSi undergoes large stress during electrochemical cycles leading to a sudden increase in interfacial resistances of charge transfer and solid-electrolyte interphase (SEI) at the end of delithiation. We further show that the rippling phenomenon also occurs in the liquid electrolyte system through galvanostatic cycles by ex situ analysis. The carbon coated 2D Si presents a new design paradigm toward mechanically durable and flexible anode materials for high-performance LIBs.

## Results

**Structural and electrochemical properties.** 2DSi was prepared via inorganic template-assisted CVD process of silane (SiH$_4$) and subsequently coated with amorphous carbon layers to form 2DSi@C (Supplementary Note 1 and Supplementary Figs. 1 and 2). Under a scanning electron microscope (SEM), the prepared 2DSi@C exhibits flat film morphology with a smooth surface and straight edges (Fig. 1a). The dimension of 2DSi was controlled to 50 nm in thickness with a micrometer size in lateral directions, in order to minimize the possibility of cracking and pulverization during lithiation (Supplementary Note 2 and Supplementary Figs. 3 and 4)[37]. Generally, the CVD process of silane generates amorphous Si (a-Si). Subsequent high temperature annealing for carbon coating transforms a-Si into polycrystalline Si, as corroborated by selected area electron diffraction (SAED) pattern, Raman blueshift and emerged peaks in X-ray diffraction in inset of Fig. 1b and Supplementary Fig. 5. TEM images confirm that an amorphous 5–10 nm thick carbon layer conformally covers the 2DSi surface, as shown in Fig. 1b, c.

The effect of carbon coating layers on the electrochemical performances of 2DSi is shown in Fig. 1d, e (See Methods). Compared to the commercially available Si nanoparticles (SiNPs) on the same scale, both 2DSi and 2DSi@C have a non-porous structure with a surface area of 10.79 m$^2$ g$^{-1}$ and 9.81 m$^2$ g$^{-1}$, respectively. This prevents excessive formation of the SEI layer and irreversible capacity loss in the first cycle, as shown in Fig. 1d and Supplementary Fig. 6. The 2DSi and 2DSi@C electrodes achieved initial reversible capacities of 2943 mA h g$^{-1}$ and 2255 mA h g$^{-1}$, corresponding to high ICE of 87.2% and 92.3%, respectively, while their redox pairs and potentials have no difference (Supplementary Fig. 7). The improved Coulombic efficiency is attributable to the increased electrical conductivity and the formation of stable SEI layer[38]. Figure 1d compares the cycling stability of 2DSi and 2DSi@C electrodes for 200 cycles at 0.2 C-rate. As previously reported in Si/C composite electrodes, carbon conductive layers buffer the large expansion of Si to a certain extent, which explains that capacity retention improved to 94.9% for the 2DSi@C electrode from 54.3% for the bare 2DSi electrode. In addition to cycling stability, structural advantage of 2DSi@C enables fast ionic transport at 20 C-rate, showing 73.9% of retention compared to initial capacity at 0.2 C-rate, which can be further elucidated by comparing the changes in their charge transfer resistance after cycles (Fig. 1f and Supplementary Figs. 8 and 9). Remarkably, 2DSi@C electrode still delivers 1145 mA h g$^{-1}$ capacity after 500 cycles at 1 C-rate, while bare 2DSi electrode dramatically decayed during the fast cycling (Fig. 1g). When pairing with conventional lithium cobalt oxide (LiCoO$_2$) cathode, its full cell demonstrated a high reversibility at initial cycle as well as extended cycle life with a high capacity loading of 3.16 mAh cm$^{-2}$ at 0.5 C-rate (Fig. 1h and Supplementary Fig. 10). While it is beyond doubt that the carbon coating layer improves electrochemical performances, it remains unclear how efficiently 2D structure can relax the internal stress to achieve long cyclability.

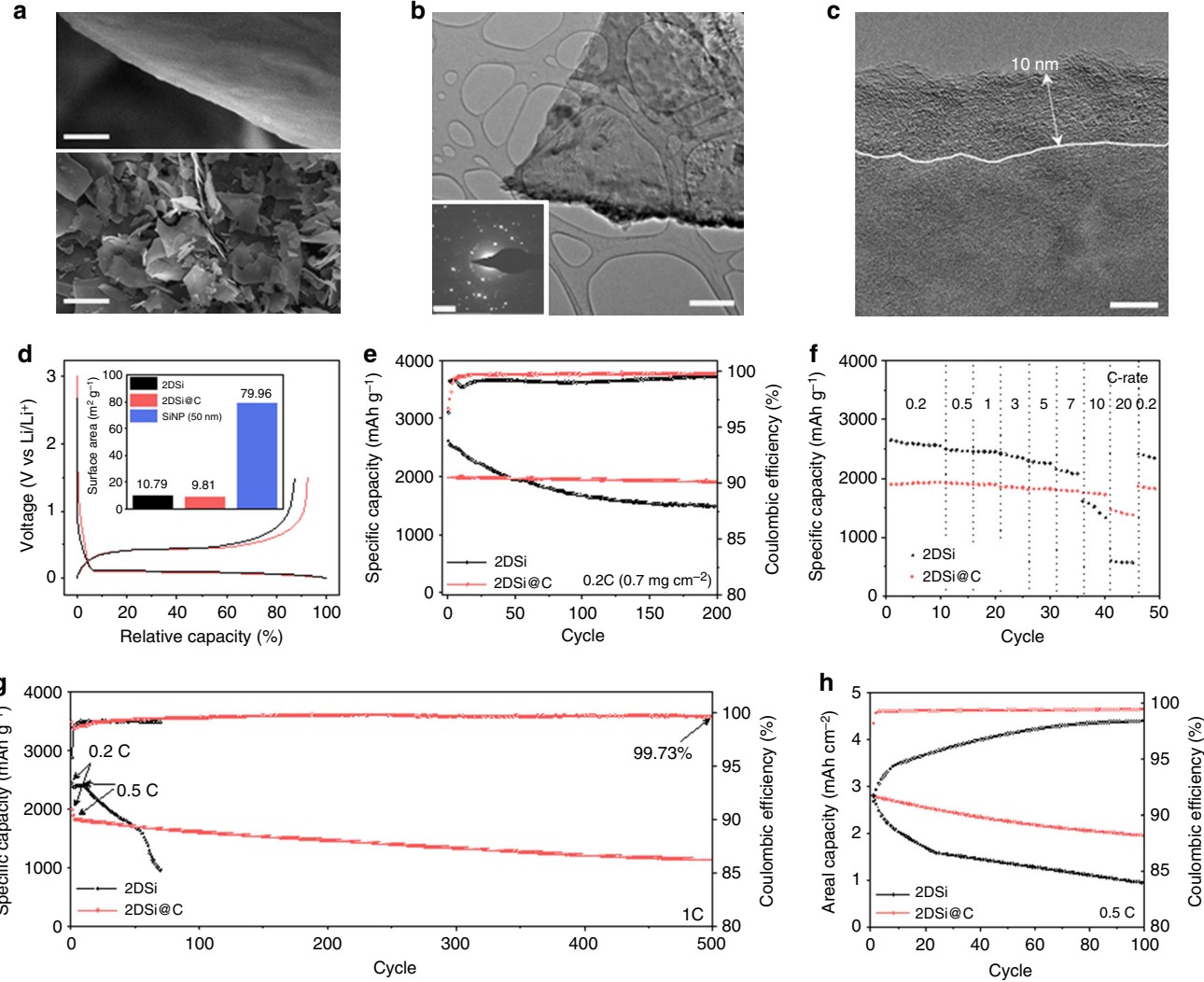

**Fig. 1** Physical and electrochemical characterization of 2DSi-based anodes. **a** SEM images of 2DSi with high and low magnifications. **b** A TEM image of 2DSi@C, inset shows the typical SAED patterns of polycrystalline Si. **c** A high magnification TEM image, showing 5–10 nm amorphous carbon layers coated on the 2DSi. **d**–**h** The side-by-side comparison of half-cell electrochemical performance between 2DSi and 2DSi@C electrodes on initial galvanostatic voltage profiles (**d**, inset: surface area results of 2DSi, 2DSi@C, and SiNP), capacity retention at 0.2 C-rate (**e**), rate capability at different C-rate from 0.2 C to 20 C for each 5 cycles (**f**), long-term stability at 1 C-rate (**g**), and capacity retention of full-cell paired with LiCoO₂ cathode (**h**), respectively. Scale bars, 100 nm and 5 μm (**a**); 500 nm (Inset: 2 1/nm) (**b**); and 5 nm (**c**)

**In situ TEM observation and chemomechanical modeling.** The structural deformation of the 2DSi@C materials in the first two lithiation/delithiation cycles were monitored by in situ TEM under a proper bias of −3V and 3 V, respectively (Fig. 2a–e and Supplementary Fig. 11 and see Methods). A 2DSi@C flake was mounted on a platinum (Pt) wire, which was connected to the Li₂O/Li probe, where the Li metal serves as the counter electrode and the native Li₂O layer as the solid electrolyte (Supplementary Movies 1–3). During the first lithiation, the flake underwent anisotropic swelling, where the lateral dimension of the fully lithiated flake expanded only 9.7%, as shown in Fig. 2b. Given that fully lithiated Si (Li₁₅Si₄ phase in SAED pattern) undergoes ~270% volumetric swelling, this implies that swelling in the thickness direction of the 2DSi@C flake should be ~200%, much larger than in the lateral direction. Interestingly, the delithiated 2DSi@C flake rippled along the lithiation direction from the near edge of 2DSi@C to the Li/Li₂O end, and overall the flake was significantly buckled, as shown in Fig. 2c. The second lithiation smoothed out the ripples and the flake looked folded due to the spatial constraint in the nanobattery system (Fig. 2d).

The delithiated 2DSi@C in the second cycle was severely buckled, while maintaining its 2D morphology without fracture (Fig. 2e).

In contrast, bare 2DSi without carbon coating layers swelled into the lateral direction as much as 31.9% after the first lithiation, while average lateral expansion in other control experiment was ~50%. This experimental variation may arise from different experimental conditions (Supplementary Fig. 12). Lithiation of the 2DSi flakes also took longer times and required higher bias than that of the 2DSi@C flakes, indicating the relatively poor rate capability. During delithiation, the 2DSi flakes could not recover its original dimensions, but preserved the swollen and sharp edges generated during lithiation. The distinctly different swelling morphologies during the cycling between the 2DSi and 2DSi@C flakes suggest the unique role of the carbon coating layers on the deformation mechanisms. Despite of the stark contrast in the swelling morphologies, we observed that Li-ion transport pathways during lithiation of these two different 2D flakes are rather in common: Li-ions diffuse fast along the edges of the flakes, followed by subsequent bulk

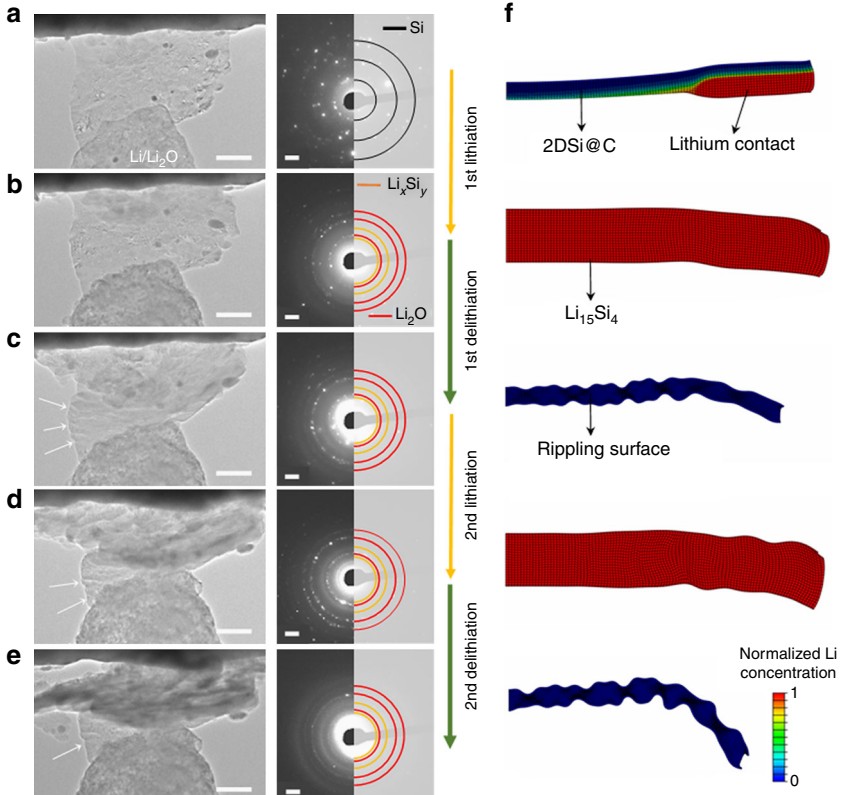

**Fig. 2** Electrochemical and chemomechanical behavior of the 2DSi@C. **a–e** The time-lapse in situ TEM images and corresponding SAED patterns of 2DSi@C for two cycles of lithiation and delithation. **f** Snapshots of chemomechanical modeling of 2DSi@C corresponding to the in situ TEM results. Scale bars, 500 nm for TEM images and 2 1/nm for SAED patterns (**a–e**)

lithiation, as evidenced by the SAED patterns of multiple spots (Supplementary Figs. 13, 14).

In order to unveil the mechanisms underlying the rippling morphology during delithiation of the 2DSi@C flakes, we developed a chemo-mechanical model to simulate the lithiation/delithiation cycles of the 2DSi@C fakes (Fig. 2f). The model accounts for both chemical and elasto-plastic deformations and couples lithiation kinetics and mechanical stress generation in a unified framework with appropriately chosen materials properties (see Methods). The cross-sectional views of the lithiation/delithiation induced deformation morphologies are shown in Fig. 2f, and the corresponding overall side views are shown in Supplementary Fig. 15. In our simulations, lithium source approached from the bottom of the 2DSi@C flake, to be consistent with the experimental settings. Owing to the much faster surface diffusivity than the bulk, Li-ions started to form the Li-Si alloys first on the surface and continued the lithiation deep inside the sheet. Consistent with the in situ TEM results, rippling occurred during delithiation along with a shortened lateral dimension. Upon full delithiation, the flake buckled on the specimen level. During the second cycle of lithiation, the large volumetric expansion diminished the rippling morphology, despite that some still remained near the lithium source, also consistent with the direct TEM observations. This repeated deformation pattern became much obvious in the second cycle of delithiation.

**Mechanical mismatch induced anisotropic expansion**. The ripple morphology of the delithiated 2DSi@C flakes was further examined by ex situ TEM (Fig. 3a) in dark-gray contrast. The ex situ study demonstrates that the ripple formation is not limited to in situ systems using $Li_2O$ as the solid electrolyte, but also

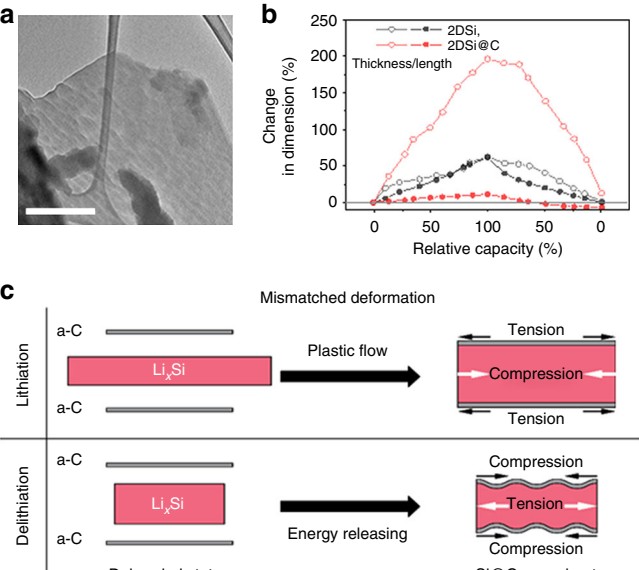

**Fig. 3** Large deswelling ratio induced rippling in 2DSi@C. **a** An ex situ TEM image of 2DSi@C after the first cycle. **b** The change in dimension versus SOC calculated from the chemomechanical modeling. **c** Illustrations of mechanical mismatch induced internal stress and rippling morphology in 2DSi@C during lithiation/delithiation process. Scale bar = 500 nm (**a**)

occurs in the liquid electrolyte systems. As directly measuring the thickness and width of a single flake in the rippled morphology is challenging, we rationalize the dimension changes from a theoretical point of view, as shown in Fig. 3b. Unlike the bare 2DSi

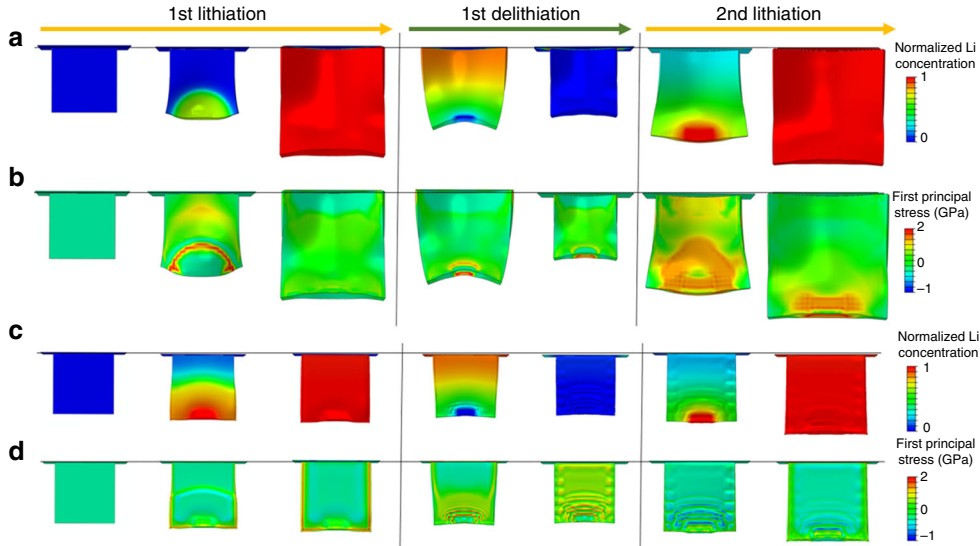

**Fig. 4** Comparisons of the morphological evolution between 2DSi and 2DSi@C by the chemomechanical modeling. Snapshots of deformation morphologies predicted by the chemomechanical model, showing **a–c** lithium concentration and **b–d** the first principal stress of 2DSi and 2DSi@C, respectively

flakes that undergo isotropic expansion and shrinkage during electrochemical cycling, the 2DSi@C flakes swelled anisotropically due to the lateral confinement by the carbon layers, as explained below. During lithiation, the polycrystalline Si undergoes isotropic expansion at the atomic scale[39], which is ~55% in each orthotropic direction (corresponding to ~270% volume increase). However, since the carbon layer deforms negligibly upon lithiation compared to Si, the carbon coating layer strongly constrains the lateral expansion of the 2DSi flake. Lithiation induced expansion in the thickness direction is, however, free of any constraints. This leads to strong anisotropic swelling in the fully lithiated 2DSi@C flakes, with only ~9.7% expansion in the lateral directions but ~200% in the thickness direction. Owing to the constraining effect of the carbon layer, large compressive stress is generated inside the lithiated 2DSi, which suppresses crack formation and propagation, as shown in Fig. 3c. In contrast, in bare nanostructured Si materials, large tensile stress is generated at the outer surface, causing surface fracture and pulverization[40,41]. The presence of the coating layer thus converts the tensile stress in the bare Si counterparts into compressive stress in 2DSi@C flakes during the lithiation process, preventing pulverization of the Si nanosheet. The compressive stress may cause lithiation retardation and possibly makes the inner regions of the 2DSi@C flake electrochemically inaccessible at the applied bias, which explains the slightly lower specific capacity of 2DSi@C compared to the bare 2DSi (Fig. 1d)[42,43]. Importantly, the interlayer strength between the amorphous carbon layer and the Si nanosheet is sufficiently strong to withstand the interlayer shearing induced by the mismatch strains, as delamination did not occur upon lithiation, as shown in Supplementary Fig. 16.

With the large disparity in the elastic modulus between carbon coating layer and lithiated product ($Li_xSi$), the compressed lithiated Si is prone to plastic yielding (the yield strength, $\sigma_Y = 1$ GPa). The plastic flow together with the in-plane confinement of the carbon layer contributes to the overall anisotropic expansion of the 2DSi@C flake upon lithiation. During delithiation, $Li_xSi$ tends to shrink isotropically (~55% in all the directions) because of its amorphous structure. However, such a large shrinkage during delithiation cannot be compensated by the expansion during the lithiation stage, recalling that the in-plane expansion upon lithiation is only ~10% in the lateral direction. This mismatched deformation between the 2D Si and its coating layer leads to the tension in the former and compression in the latter. To relieve the excessive compressive energy, the carbon layer ripples[44]. The rippling morphology not only reduces the compressive stress in the coating layer and but also the tension in the delithiated Si, which renders the 2D Si mechanically more durable.

**Stress distribution and interfacial resistance in rippled structure.** To further elucidate the rippling mechanisms described above, we simulated the lithiation/delithiation in the 2DSi@C flakes in comparison with bare 2DSi flakes (Supplementary Fig. 17)[11,45], as shown in Fig. 4. Our simulations show that the first principal stress is much higher in 2DSi than in 2DSi@C during lithiation. This stress difference becomes more pronounced from the second cycle. The stress analysis manifests that 2DSi is prone to crack nucleation and growth at the edges in the second lithiation than in the first lithiation, though its thickness is less than the critical size (~150 nm) of Si nanomaterials[37]. Indeed, from the in situ observations on bare 2DSi with dimension over 2 μm in the lateral directions, cracks were initiated from the lithiation front and propagated to the counter electrodes (Supplementary Movies 4, 5). This fracture phenomenon of nano-thick 2DSi flakes suggests that crack formation also depends on the lateral dimension of 2D materials, as shown in Supplementary Fig. 18. In contrast, the 2DSi@C layer was able to release the stress through the formation of ripples and had a quite resilient structure upon the second lithiation, as shown in Fig. 4b, d.

The ripple formation and accompanying stress-releasing ability significantly affect the interfacial resistances of charge transfer and SEI layers. To investigate the correlation between resistances and stress distributions in the 2D Si materials, in situ galvanostatic electrochemical impedance spectroscopy (EIS) measurements were conducted, as shown in Supplementary Figs. 19 and 20[46,47]. Unlike conventional potentiostatic EIS, galvanostatic measurement of in situ EIS can define the rate-dependent kinetics of electrode by applying the sinusoidal current signal. Typical 'w' pattern indicates that lithiation of Si is obviously shown as a function of state-of-charge (SOC). During an initial stage of SOC, the resistance gradually decreased due to the formation of crystalline lithiated Si ($Li_xSi$) that enhances charge transfer kinetics[47,48]. When Si was fully discharged to the form of $Li_{15}Si_4$,

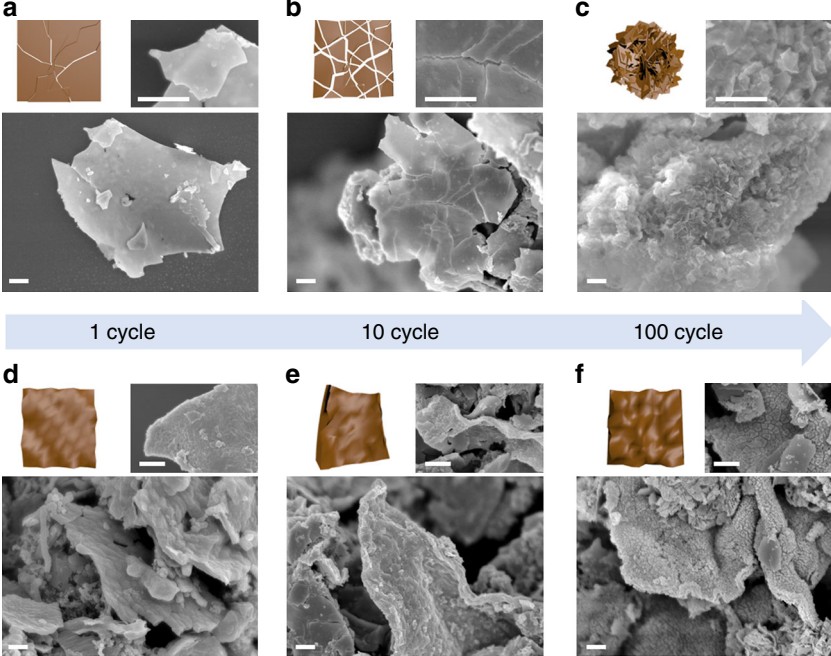

**Fig. 5** The morphology changes in a single sheet. A series of SEM images of **a–c** 2DSi and **d–f** 2DSi@C after 1, 10, and 100 cycles, respectively (Inset: schematic illustration of each morphology and high magnification SEM images). Scale bars are 1 μm

both resistances increased, because $Li_{15}Si_4$ surface obstructs additional diffusion of Li-ions. The resistance ratio of $R_{2DSi@C}/R_{2DSi}$ reached ~1.0 when fully discharged. However, during delithiation, the resistance rapidly decreased from the relative capacity of 50%, where the rippling formation was observed in earnest (Supplementary Fig. 20). The actual resistance values in 2DSi were 40 times as high as in 2DSi@C. The reason for this large difference may be attributed to the unstable SEI formation during delithiation of 2DSi. The high stress predicted in the simulations may lead to nucleation and propagation cracks in 2DSi, which may further accelerate continuous decomposition of electrolyte and thicken the SEI layers. In contrast, rippled structure in 2DSi@C can efficiently release the internal stress and improve the structural integrity, which in turn enhance the interface stability and reduce the resistances.

### Discussion

There have been numerous achievements for suppressing the volume expansion of Si nanomaterials by either introducing pre-reserved void spaces or porous structures. However, extra empty spaces greatly reduce the tap density of materials and porosity produces a thicker SEI layer with a poor reversibility in the first cycle, both hinder practical applications of Si anodes. From the in situ observations and theoretical analyses, we here demonstrated that 2D Si materials with carbon coating layers could fully address the aforementioned issues and bear the stress-releasing ability during cycling. This will enhance the durability and cyclability of the 2DSi anodes (Supplementary Table 1).

The morphological evolutions of both 2DSi and 2DSi@C show a stark difference in subsequent cycles. As expected from both the in situ EIS results, cracks began to form from the edges of the 2DSi sheets after the first cycle and propagated rapidly over the sheets. Through the newly exposed fresh surface of Si, electrolytes were continuously decomposed, leading to a typical failure mode of Si anodes. As a result, the 2DSi broke into smaller pieces and largely agglomerated, as shown in Fig. 5a–c. In contrast, during lithiation the Si nanosheet in 2DSi@C undergoes

compression and during delithiation the 2DSi@C forms ripple morphology with a highly extended degree of wrinkles that release internal stress in the Si nanosheets. Both features suppress fracture of the 2DSi@C during the electrochemical cycling. In our chemomechanical modeling, delithiation should proceed from the lithium contact site and the uniaxial Li-ion flow produces relatively regular ripples on the surface, while ex situ analysis showed rather randomly oriented ripples after cycles and relatively small bending because of complex structure of composite electrodes involving neighboring particles. This causes an island-like surface morphology after 100 cycles, which enables the stable battery operation during repeated cycles by the formation of stress-relaxation rippling structure, as shown in Fig. 5d–f.

Besides, close relationship between the coating thickness and rippling structure through a finite element analysis was investigated (Supplementary Note 3 and Supplementary Fig. 21). The thicker coating layers is, the less favorable rippling structure is, while a thin coating layer produces the ripples with large amplitude. As the coating thickness increases to a certain point, in-plane compressive energy of coating layer cannot be relaxed. In this respect, we made an attempt to explore optimum thickness of the coating layers estimated by comparing the in-plane stress which mainly occurs on coating layers during lithiation and on Si nanosheet during delithiation (Supplementary Note 3 and Supplementary Fig. 22). For the thinner coating layers, the in-plane stress will exceed the fracture strength of the coating and possibly cause the cracking in the coating layer during lithiation. The thicker layers lead to smaller rippling amplitude, which causes very large in-plane tensile stress inside the Si nanosheet. While the above scaling analysis shows that an optimal thickness regime exists where both the coating layer and the Si nanosheet will not fracture, further mechanistic implications in rippling structure needs future studies.

The present in situ observations corroborated with mechanics analysis have revealed unusual swelling/deswelling mechanisms as well as stress-releasing ability in carbon-coated 2D Si anodes during electrochemical cycling. During lithiation carbon coating on the 2D Si acts as a protective layer that suppresses crack

nucleation and propagation. During subsequent delithiation the large deswelling ratio of Si and carbon induces ripple formation. The ripple morphology has stress resilient characteristics in repeated cycles and facilitates maintaining structural integrity and interfacial stability, as demonstrated by in situ EIS measurements. In addition to the fundamental understanding of the chemo-mechanical degradation mechanisms in the 2D Si anodes, the present study opens a new route toward high-performance flexible batteries to power wearable electronics.

## Methods

**Preparation of 2D Si**. Silane gas with six nine (99.9999%) purity was thermally decomposed in a rotary furnace which contains NaCl as a template. For a production of 50-nm-thick Si onto the template, the CVD process was conducted at 550 °C with a flow rate of silane gas (50 sccm). After pyrolysis of silane, NaCl was selectively removed by intense stirring in DI water and subsequently 2D Si with each thickness was obtained after filtration and drying steps. The resulting 2DSi sheet was uniformly coated by amorphous carbon by decomposition of acetylene gas ($C_2H_2$) at 900 °C for 5 min, which produced 7 wt% of carbon in the 2DSi@C.

**Material characterization**. Microstructural evolution was investigated using SEM (Verios 460, FEI), TEM (JEM-2100F, JEOL), Raman (NRS-3000, JASCO spectrometer), XRD (Bruker D8-Advance), and Brunauer–Emmett–Teller analysis (BET, ASAP2020, Micromeritics Instruments).

**Electrochemical characterization**. Electrochemical test was evaluated using 2032 coin-type cell. The working electrode was composed of 80 wt% as an active material, 10 wt% binder ((poly(acrylic acid)/sodium carboxymethyl cellulose, PAA/CMC, 50 wt%/50 wt%, Sigma-Aldrich)) and 10 wt% super-P (TIMCAL), which were prepared by slurry coating method. The loading levels of series of 2DSi was 0.5–1.1 mg cm$^{-2}$. The coin cells, which composed of 2D Si electrodes, polymer separator (Celgard 2400), and a lithium counter electrode were assembled in an argon-filled glove box. The electrolyte was a 1.3 M LiPF$_6$ solution in ethylene carbonate/diethyl carbonate (3/7 vol%) with 10 wt% fluorinated ethylene carbonate additives included to improve the cycling stability. All the specific capacities described above were calculated based on the weight of active materials. In particular, specific capacity of 2DSi@C were calculated by including the weight of carbon coating layers. For cathode, LiCoO$_2$ (LCO, L&F Inc.) powder was used as active material with its composition 95:2:3 (LCO: polyvinylidene fluoride (PVdF): super-P) and with a loading level of ~20 mg cm$^{-2}$. For full cell, theoretical N/P ratio was ~1.1. The operating voltage window was from 0.005-1.5 V at a 0.05-20 C-rate for anode half-cell, 3.0–4.3 V for cathode half-cell, and 2.5–4.2 V for full-cell. Electrochemical properties were measured using a cycle tester (WBCS3000 battery systems, Wonatech).

**In situ EIS**. A set of multiple impedance spectra were measured every 30 min by the galvanostatic EIS at a constant current mode (0.1 C) during lithiation/delithiation, since the potantiostat of in situ EIS system consisted of two different channels: one was for measuring impedance spectra, the other was for recording voltage profiles. Input signals were generated by the superposition of sinusoidal current waves of 10 mA amplitude at 200 kHz to 1 Hz (VSP-300, BioLogic).

**In situ TEM measurements**. In situ TEM measurements were carried out with a Nanofactory TEM-STM holder inside a Titan 80–300 scanning/transmission electron microscope (S/TEM). One probe consisted of 2DSi samples which were drop-cast onto a gold wire as a working electrode, and the other probe was a tungsten probe with a piece of Li metal attached to the tip as a counter electrode as shown in Supplementary Fig. 23. The probes were affixed to the TEM holder in an Ar-filled glovebox and transported to the TEM in an airtight container, where the holder was then removed and inserted into the TEM. A native Li$_2$O layer is formed by a short exposure (3–5 s) of Li to air functions as a solid-state electrolyte[37,49]. Inside the TEM, a piezo-positioner is used to move the Li/Li$_2$O electrode into contact with the 2DSi samples. After contact, a bias of –3.0 V was applied to initiate lithiation. Once lithiation was completed, a bias of 3 V was applied for delithiation. In order to drive the Li$^+$ ions through the solid electrolytes, applied potentials are larger than those used in real Li–Si cell. However, this is common for in situ TEM studies; for example, in situ lithiation of Si requires the application of –2.0 V vs Li electrode, whereas the lithiation window in Li–Si half cells is 0 to +2.0 V vs Li/Li$_2$O. This difference does not appear to change the behavior of the material[7,49].

**Chemomechanical modeling**. We employ a recently developed chemo-mechanical model to simulate the lithiation/delithiation process in bare Si nanosheet and 2DSi@C. For Si, in the finite-strain framework, lithiation induced deformation consists of the stretch rates and the spin rates. The total stretch rate is additive of the three components, the chemical ($\dot{\varepsilon}_{ij}^c$), elastic ($\dot{\varepsilon}_{ij}^e$), and plastic ($\dot{\varepsilon}_{ij}^p$) one, $\dot{\varepsilon}_{ij} = \dot{\varepsilon}_{ij}^c + \dot{\varepsilon}_{ij}^e + \dot{\varepsilon}_{ij}^p$. The chemical stretch rate is assumed to be proportional to the

increment of the normalized lithium concentration, $\dot{\varepsilon}_{ij}^c = \beta_{ij}\dot{c}$. The diagonal tensor, $\beta_{ij}$, represents the lithiation induced expansion. We set $\beta_{11} = \beta_{22} = \beta_{33} = 0.6$, and $\beta_{ij} = 0$ for the other entries. The elastic stretch rate, $\dot{\varepsilon}_{ij}^e$, obeys Hooke's law with the stiffness tensor, $C_{ijkl}$, depending on 2 independent material constants (i.e., Young's modulus $E$ and Poisson's ratio $\nu$). For the intermediate states of charge, the stiffness tensor is assumed to be linearly dependent on lithium concentration, interpolated by these two extreme states. The plastic stretch rate, $\dot{\varepsilon}_{ij}^p$, obeys the classic $J2$-flow rule. Namely, plastic yielding occurs when the equivalent stress, $\sigma_{eq} = (3s_{ij} s_{ij}/2)^{1/2}$, reaches the yield strength. Here $s_{ij} = \sigma_{ij}-\sigma_{kk}\delta_{ij}/3$ is the deviatoric part of Cauchy stress, $\sigma_{ij}$, and $\dot{\varepsilon}_{ij}^p$ is proportional to $s_{ij}$. Note that we assume that both chemical and plastic deformations are spin-free. For carbon coating layer, we ignore the changes caused by lithiation/delithiation, and assume it deforms elastically.

In lithiation in crystalline Si (2DSi@C) and the first-step lithiation in amorphous Si (bare 2DSi), a critical feature is the formation of sharp interphase that separates the Li-poor and Li-rich domains. To capture this feature, lithium diffusivity is set to be nonlinearly dependent on lithium concentration: $D = D_0[1/(1 − c) − 2\alpha c]$, where $D_0$ and $\alpha$ are tunable constants to control the interphase profile between the Li-rich and Li-poor. The two-step lithiation in amorphous Si is realized by setting different Dirichlet boundary conditions (i.e., $c = 0.67$ for Li$_{2.5}$Si and $c = 1$ for Li$_{3.75}$Si).

This chemo-mechanical model is numerically implemented in the finite element package ABAQUS/standard, with user subroutine UMATHT to realize the sharp interphase. For pure amorphous Si, we set $E_{aSi} = 100$ GPa and $\nu_{aSi} = 0.22$, while for pure crystalline Si $E_{cSi} = 160$ GPa and $\nu_{cSi} = 0.22$. For fully lithiated Si, we set $E'_{Si} = 25$ GPa and $\nu_{Si} = 0.22$. Yield strength of Si is set to be 1 GPa. For carbon coating layer, we set $E_c = 250$ GPa, $\nu_c = 0.25$ and $\sigma_{cY} = 30$ GPa.

**Data availability**. The authors declare that the data supporting the findings of this study are available within the article and its Supplementary Information files. All other relevant data supporting the findings of this study are available on request.

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

## Acknowledgements

This work was supported by the Center for Advanced Soft-Electronics funded by the Ministry of Science, ICT and Future Planning as Global Frontier Project (CASE-2015M3A6A5072945), MOE(BK21Plus:META, 10Z20130011057) and LG Younam Foundation. C.M.W. thanks the support of the Assistant Secretary for Energy Efficiency and Renewable Energy, Office of Vehicle Technologies of the U. S. Department of Energy under Contract No. DE-AC02-05CH11231, Subcontract Nos. 18769 and 6951379 under the Advanced Battery Materials Research (BMR) program. The microscopic analysis in this work was conducted in the William R. Wiley Environmental Molecular Sciences Laboratory (EMSL), a national scientific user facility sponsored by DOE's Office of Biological and Environmental Research and located at PNNL. PNNL is operated by Battelle for the Department of Energy under Contract DE-AC05-76RLO1830. S.Z. acknowledges the support by the National Science Foundation through the projects CMMI-0900692, DMR-1610430, and ECCS-1610331.

## Author contributions

J.R., T.C., and T.B. contributed equally to this work. S.P. conceived the concept. T.B. designed and carried out the experiments, physical characterization and electrochemical test. J.R., G.S., L.L., and C.W. performed the ex situ and in situ TEM characterization and data analysis. T.C. and S.Z. designed the modeling and simulations. J.M. and J.C. performed the CVD of silane. C.H. and H.S. performed in situ EIS and data analysis. J.R., T.C., T.B., and S.Z. wrote the manuscript. All authors discussed the results and commented on the manuscript.

## Additional information

**Competing interests:** The authors declare no competing interests.

