## [Peer Review File · Nature Communications]

Reviewers' comments:

Reviewer #1 (Remarks to the Author):

This work introduces a 2D Si nanosheet coated with a carbon layer as anode material for LIBs and characterizes the deformation process during lithiation and delithiation cycles by in-situ TEM. This paper is not suggested to be published in Nature Communications. The performance of the 2DSi@C is not high and the synthesis and characterizations is lack of novelty.

1. The cycling performance of the 2DSi@C is poor. The authors only provided the data for 200 cycles, and the current density was low.
2. As for the rate performance, Fig. 1f only gives the information about the capacity retention at various current densities, which is not enough to present the rate performance. More datum in detail are needed.
3. The average coulombic efficiency of the 2DSi@C half battery seems to be not very high, and even lower than the bare Si nanosheet half battery after the 150th cycle. The low CE indicates that the ripple morphology may not be able to facilitate maintaining structural integrity and interfacial stability after long cycles.
4. The EIS results of both 2DSi@C half batteries and the control batteries after 100 cycles and 200 cycles should be provided to show the structural stability of the 2DSi@C electrodes.

Reviewer #2 (Remarks to the Author):

This paper introduces a design of silicon anode that has a thin carbon layer coated on a 2D Si nano sheet, and demonstrated an increased cycling stability compared to the uncoated 2D Si nano sheet. Due to differences in the material properties of silicon and carbon, the proposed design forms a rippling structure during cycling which releases some of the stress. The authors provide evidence from in-situ TEM as well as necessary electrochemical tests, and use a chemomechanical model to support their observations in TEM.

Overall, the manuscript is well written with no obvious flaws. The abstract and conclusions are both clear and legitimate. The electrochemical data is well presented. There are a number of interesting work on nano-engineering of silicon anodes, and the rippling structure and its stress releasing mechanisms are new.

This paper contributes to the development of new anode materials that undergo large volumetric change, but the applicability and significance to other disciplines needs better justifications. For instance, how to take advantage of the large deformation materials and how does this idea inspire work in other disciplines? How is the practicability of this method and its performance compared to the existing nano-engineered Si anode structures?

Though a numerical simulation is used to help explain the stress resilient characteristic, a deeper understanding can strengthen this paper even more. For instance, does the rippling behavior depend on the thickness of the 2D Si nano sheet? In what situations will this formation of the rippling structure break down?

Reviewer #3 (Remarks to the Author):

The large volume change of Si electrode upon electrochemical cycling is always a critical challenge for a stable Li-ion battery. In this paper, authors introduce a novel battery designing concept taking advantage of mechanical mismatch-driven rippling in carbon-coated Si sheets. The mechanism of the deformation patterns is demonstrated by in-situ TEM observation and

chemomechanical simulation. Importantly, the ripple formation can release the cycling induced stress, and might help improve the electrochemical performance of Li-ion battery. However, there are some important issues that the authors have to address before this paper is accepted for publication in Nature Communications.

1. In this paper, 2D Si sheet is coated by an amorphous 5-10 nm thick carbon layer. The author found that this carbon coating can well restrict the huge expansion of 2D Si sheet upon the lithiation and form a rippling upon delithiation. The authors have to also confirm that the carbon coating at this thickness would not self-break or delaminate when it is restricting the huge expansion or forming a rippling, using either in-situ or numerical methods.

2. The thickness of carbon coating is a critical factor to affect the ripple formation and stress release within the electrode. Is it possible to find out an optimized thickness of carbon coating by numerical simulation?

3. In the Fig. 3(c), we can find the carbon coating will induce a field of tension stress within the 2DSi@C upon delithiation, while, for 2DSi sheet without carbon coating, it is basically stress-free state. It can be also confirmed by Fig. 4 (the 5th plots in 2nd and 4th rows). The question is: will this high tensile stress generate the cracks within the Si sheet? The authors should have a discussion about this tensile stress developed in 2DSi@C.

4. In this paper, the simulation well explains the experimental observations. However, the details about the modelling are missing. I would recommend that one more section in "Method" discussing about the numerical modelling would make the simulation result more convincing.

5. Since there is no discussion about the numerical modelling, I cannot see if the modelling of the lithiation/delithiation of the Si electrode is appropriate or not. Usually, the modellings of the lithiation/delithiation of the amorphous Si and polycrystalline Si electrodes should be different (2012, H. Yang, Nano letters, 12 (4), 1953 & 2016, R. Xu, Extreme Mechanics Letters, 8, 13-21 & 2015, C.V. Di Leo International Journal of Solids and Structures 67 283-296). Since the 2DSi is amorphous Si and 2DSi@C contains polycrystalline Si, has the difference between lithiations of the amorphous Si and polycrystalline Si been considered during the modelling?

6. In Page 5 Line 124: What does the "40Ag-1" correspond to the Crate? Since the authors used the Crate to describe the cyclic rate in the rest of the manuscript, it is better to also use the Crate here.

7. Are the specific capacities of 2DSi and 2DSi@C calculated by the weight of active materials? For the specific capacity of 2DSi@C, is the weight of carbon coating taken into account? The authors should clarify this.

8. The authors claim the ripple formation also occurs in the liquid electrolyte system. What is the electrolyte the authors used in this paper? Were any additives added into the electrolyte? It is very important because the electrolyte would significantly change the electrochemical performance of the Si electrode. The authors should add more details and necessary information in the electrochemical characterization section to make the experiments more convincing.

9. In the section from Line 229 to Line 244, the authors misused the "lithiation" and "charge". For example, Line 234, "the Si was fully charged to the form of $\text{Li}_{15}\text{Si}_4$ " should be "the Si was fully discharged to the form of $\text{Li}_{15}\text{Si}_4$ ". Same misuses can be found elsewhere in this section. The authors have to modify this.

10. Please add references for the discussion in Line 234 "the resistance gradually decreased due to the formation of crystalline lithiated Si (Li_xSi) that enhances charge transfer kinetics"

Responses to all reviewer comments for Manuscript ID: NCOMMS-18-01149

*Blue: Words in the original paper, Red: Modified version.

Reviewer #1 (Remarks to the Author):

This work introduces a 2D Si nanosheet coated with a carbon layer as anode material for LIBs and characterizes the deformation process during lithiation and delithiation cycles by in-situ TEM. This paper is not suggested to be published in Nature Communications. The performance of the 2DSi@C is not high, and the synthesis and characterizations is lack of novelty.

Response: We appreciate the reviewer's helpful comments. First of all, we hope that the reviewer understands that this original paper aims at reporting, for the first time, new observations on the evolution of stress-releasing structure of 2D Si nanosheet coated with thin amorphous carbon layers along with suitable validation from chemomechanical modeling. As we focused on disseminating the new findings as the purpose of this paper with condensing huge amounts of data, the reviewer might feel that several electrochemical figure-of-merit are not satisfactory enough to be published in *Nature Communications*. With this opportunity to further improve the quality of our submitted paper, we significantly supplemented the electrochemical performances as the reviewer pointed out and reorganized those data as well as several unsuitable results as follows.

The synthetic procedure for preparing the 2DSi nanosheets is facilitated by recyclable salt templating method and a large-scale CVD process. Recently reported 2DSi materials have ultrathin and porous structure fabricated by molten-salt induced exfoliation, exfoliation of lithium silicide, graphene oxide templating, and reduction from mesoporous SiO₂. These products are not suitable for battery-grade material design, due to the large surface area and very low tap density, even though they showed decent electrochemical results. Large surface area accelerates a large amount of electrolyte decomposition, leading to a poor initial Coulombic efficiency (<80%) and electrode density or energy density of LIBs have been greatly hindered by low tap density of nanostructure materials. In this light, we introduced the large-scale CVD process to produce non-porous 2D Si materials on highly recyclable inorganic template (>99%) where the single experimental batch exceeds 500 g. We applied non-solvent induced recrystallization method for designing the template size in suitable range (<20 μm) for active materials as well as for high recycling efficiency. Compared to previous approaches, our proposed method can provide rather scalable direction to prepare nano-thick 2DSi materials with non-porous features. **Supplementary Note 1** along with **Supplementary Fig. 1** describes novelty in our synthesis method in detail.

Supplementary Figure 1. Synthesis of 2DSi anode via recyclable salt templating method.

(a) Schematic illustration showing synthetic process of 2DSi. (b) SEM images of 10 μm -sized cubic crystal of NaCl. (c) SEM image and the corresponding EDS image of Si@NaCl. (d) Recycling efficiency for 10 cycles and the amount of salt recycled starting from initial amount of 500 g. (e) Particle size distribution of 2DSi materials based on the SEM images. Scale bars are 100 μm (inset: 5 μm) and 5 μm for (b) and (c), respectively.

Furthermore, the thickness of 2DSi nanosheets can be readily controllable through changing the decomposition time of silane gas. As shown in **Supplementary Fig. 2**, 50 nm to 380 nm-thick 2DSi materials are prepared without morphological variations. Thus, we could rationalize our choice on 50 nm-thick 2DSi for further studies like in-situ TEM or simulation works, since 50 nm-thick 2DSi anodes have shown the most promising results on every

electrochemical aspect such as ICE, retentions, rate capability and kinetics (**Supplementary Figs. 3, 4**). **Supplementary Note 2** explains thickness-dependent electrochemical properties of 2DSi anodes in detail. Besides, variation of coating thickness cannot be easily achievable in rather processible way, since typical carbon coating through CVD process has a limitation to generate over than 20 nm-thick amorphous carbon on target materials. If we try to make the samples with thicker carbon layers, isolated carbon clusters or discontinuous carbon layers are formed. Therefore, this aspect can be only validated by simulations.

Supplementary Figure 2. Variable Thickness of 2DSi. SEM images of (a) 50 nm-thick, (b) 100 nm-thick, (c) 150 nm-thick, and (d) 380 nm-thick 2DSi nanosheets. The scale bars are 500 nm for left column and 10 μm for right column.

Supplementary Figure 3. Characterization of various 2DSi@C electrodes. (a) CVD reaction time-dependent thickness of 2DSi and their surface areas. (b) BJH pore size distribution curves for 2DSi with different thickness. Thickness-dependent properties: (c) Charge capacities and ICE values, (d) cycle retention at 0.2 C-rate with Coulombic efficiency, (e) charge capacities at different C-rate, and (f) electroactive-surface-area calculation plots based on CV results.

Supplementary Figure 4. CV results of 2DSi electrodes with different thickness. CV curves with a variation of scan rate from 0.2 to 1.0 mV s^{-1} of (a) 50 nm, (b) 100 nm, (c) 150 nm, and (d) 380 nm-thick 2DSi electrodes.

1. The cycling performance of the 2DSi@C is poor. The authors only provided the data for 200 cycles, and the current density was low.

Response: Since the presented electrochemical results are based on the high mass electrode ($> 1 \text{ mg cm}^{-2}$, not compositing with graphite or other additives), cycling at high current density or long-term evaluation can be hardly achieved even with previously reported state-of-art composite electrode. Initially, we just wanted to show a demonstration of pure 2DSi@C anodes toward high energy cells without the use of special binders or electrode architecture. Furthermore, typical Si electrodes in absence of as-built void spaces and unique coating layers (not carbon), and high proportion of active materials (80wt%) in those electrodes result in unavoidable pulverization of particle and delamination of electrode, which eventually lead to structural failure and degradation of cell properties, although Si materials are prepared in nanoscale. However, our 2DSi@C electrode (high active materials loading with conventional electrode configuration) achieved 86.8% capacity retention over 200 cycles at 0.2 C-rate by the unique stress-releasing structure during cycles, which outperformed other dimensional Si nanomaterials. As noted in the manuscript, we rather take advantage of a large deformation of Si anode to stabilize the structure and improve the overall electrochemical properties, which is further corroborated by in-situ TEM observation and simulation results. If not considering these points but only paying attention to numerical specifications, it is hard to appreciate true value of our findings.

Yet, we also conducted a cycling test on less loaded electrodes of 0.7 mg cm^{-2} for active 2DSi materials to give a better electrochemical performance as the reviewer probably want to see. Interestingly, bare 2DSi electrode showed a similar capacity decay (54.3% retention) in less loaded electrode ($\sim 0.7 \text{ mg cm}^{-2}$), while 2DSi@C electrode showed much improved capacity retention of 94.9% after 200 cycles at 0.2 C-rate. This difference suggests that bare 2DSi itself cannot maintain its original structure without carbon coating layers regardless of loading levels but evolved rippling structure in the 2DSi@C nanosheet after cycles could effectively release the generated internal stress if we exclude the several engineering factors. Furthermore, we added the long-term cycling results of both electrodes for 500 cycles at 1 C-rate and full cell results which is paired with LiCoO₂ cathode, showing practical feasibility of as-prepared 2DSi@C nanosheets. We replaced cycling data from less loaded electrodes with original one and added several electrochemical results to have a novelty in its performances in Fig. 1 of the revised manuscript as follows:

Figure 1. Physical and electrochemical characterization of 2DSi-based anodes. (a) SEM images of 2DSi with high and low magnifications. (b) A TEM image of 2DSi@C, inset shows the typical SAED patterns of polycrystalline Si. (c) A high magnification TEM image, showing 5-10 nm amorphous carbon layers coated on the 2DSi. (d-h) The side-by-side comparison of half-cell electrochemical performance between 2DSi and 2DSi@C electrodes on initial galvanostatic voltage profiles (d, inset: surface areas of 2DSi, 2DSi@C, and SiNPs),

capacity retention at 0.2 C-rate (e), rate capability (f), long-term stability at 1 C-rate (g), and capacity retention of full-cell (h), respectively. Scale bars, 100 nm and 5 μm (a); 500 nm (Inset: 2/nm) (b); and 5 nm (c).

Supplementary Figure 10. Cathode and full cell evaluation. Voltage profiles at initial cycle at 0.1 C-rate (a) and cycle retention at 0.5 C-rate for LiCoO₂ cathode (b) in the potential window of 3.0-4.3V. Voltage profiles of LiCoO₂//2DSi (c) and LiCoO₂//2DSi@C (d) full cells during cycles in the potential window of 2.5-4.2V.

2. As for the rate performance, Fig. 1f only gives the information about the capacity retention at various current densities, which is not enough to present the rate performance. More datum in detail are needed.

Response: Along with summarized capacity retention values in Fig. 1f, we added the raw data for the rate capability test in **Supplementary Fig. 8**.

Supplementary Figure 8. Rate capability results of 2DSi-based electrodes. Capacity retentions of 2DSi (black) and 2DSi@C (red) at 0.2-20 C-rate for both discharge/charge.

3. The average coulombic efficiency of the 2DSi@C half battery seems to be not very high, and even lower than the bare Si nanosheet half battery after the 150th cycle. The low CE indicates that the ripple morphology may not be able to facilitate maintaining structural integrity and interfacial stability after long cycles.

Response: As we described above, we replaced the cycling results in Fig. 1e. In the replaced results, 2DSi@C electrode has an average Coulombic efficiency of 99.70% while that of 2DSi is 99.18%, which evidently support that rippling structure in 2DSi@C anode facilitated structural integrity over 200 cycles and interfacial stability (**Supplementary Fig. 9**).

4. The EIS results of both 2DSi@C half batteries and the control batteries after 100 cycles and 200 cycles should be provided to show the structural stability of the 2DSi@C electrodes.

Response: As the authors pointed out, we have already conducted ex-situ EIS measurement and added those data in **Supplementary Fig. 9**. As shown in the below figure, the charge transfer resistance of bare 2DSi electrode continuously increased for 200 cycles, particularly

exceeding 4 times, while 2DSi@C electrode has the improved structural stability and thus lower interfacial resistance over the cycles.

Supplementary Figure 9. Ex situ EIS results. Impedance spectra of 2DSi and 2DSi@C half-cell after 1st, 100th cycles and 200th cycles.

Reviewer #2 (Remarks to the Author):

This paper introduces a design of silicon anode that has a thin carbon layer coated on a 2D Si nano sheet, and demonstrated an increased cycling stability compared to the uncoated 2D Si nano sheet. Due to differences in the material properties of silicon and carbon, the proposed design forms a rippling structure during cycling which releases some of the stress. The authors provide evidence from in-situ TEM as well as necessary electrochemical tests and use a chemomechanical model to support their observations in TEM.

Overall, the manuscript is well written with no obvious flaws. The abstract and conclusions are both clear and legitimate. The electrochemical data is well presented. There are a number of interesting work on nano-engineering of silicon anodes, and the rippling structure and its stress releasing mechanisms are new.

Response: We appreciated the reviewer's favorable and detailed appraisal of our work.

This paper contributes to the development of new anode materials that undergo large volumetric change, but the applicability and significance to other disciplines needs better justifications. For instance, how to take advantage of the large deformation materials?

Response: We appreciated the reviewer's helpful comments. As we sharpened the focus of our observations in this manuscript, we took advantage of high-volume-change materials rather than fighting against such a large deformation in most previous studies. Introducing a rigid coating layer or void spaces could address the critical issues in alloy-type anodes with high degree of lithiation as far as it goes. However, there has been few attempts to design the Si anodes in its deformed state. The large deformation materials expand over than 300% in

volume with or without anisotropic changes. When the surface constraint exists especially for 2DSi materials, structural anisotropic expansions (limited expansion to lateral direction) occur unlike anisotropy from crystallographic orientations upon lithiation. During delithiation of stress-free state, it should follow the isotropic contraction in each orthogonal direction, leading to the formation of rippled patterns on Si sheets, which can release the stress and maintain the overall morphology for many cycles. This unique phenomenon is caused by mechanical mismatch in Si and C as well as 2D structure, since we did not observe such deformations in other combinations of structures with carbon coating layers.

More specifically, in bare 2D Si nanosheet, lithiation generates tension on the surface (similarly as other nanostructures like Si nanoparticles), as shown in figures below. The tensile stress leads to surface crack formation and eventually pulverization. On the contrary, in 2DSi@C nanosheet, the constraining effect of the carbon coating layer converts the tension in bare Si to compression on the surface during lithiation, which suppresses crack formation and propagation and therefore avoids pulverization. During delithiation in 2DSi@C, tensile stress is generated in the Si nanosheet, while compression in the coating layer. However, due to the formation of ripples, significant mechanical energy is released during delithiation in 2DSi@C, and therefore tension in Si nano sheet is controlled and does not cause fracture either. As the figures shown below, measured by the first principal stress (σ_1), the regions of high tension reduced substantially, which, we believe, is the reason of the better electrochemical performance of 2DSi@C compared to bare 2DSi.

In all, the unique attribute of the 2DSi@C is that the coating alters the mechanical stress-generation pathway seen in bare Si (*i.e.*, converting tension to compression in lithiation), and releases the mechanical energy in each cycle, making it chemomechanically very durable and electrochemically superior.

Figure R1. Comparison of stress state in (a) bare 2DSi during lithiation and in (b) 2DSi@C during delithiation.

and how does this idea inspire work in other disciplines?

Response: It is obvious that this idea also works for other high-volume-change materials (*i.e.*, Ge, Sn, Sb, etc.) as long as it has a proper coating layer with a different mechanical property in the 2D structure. While we should consider other physical or electrochemical properties, sodium-ions cause a huge deformation in the above choices during sodiation/desodiation as well, which implies that they also lead to similar observations. Given that those materials

have a same degree of lithiation or sodiation (how many Li/Na are alloyed with host materials), sodium-ions should cause a larger deformation due to its larger atomic size than that of Li. This kind of stress-releasing design could make a breakthrough in developing promising anodes for sodium-ion batteries or other types of metal-ion batteries.

Other possible applications of this idea could be developed for high energy density flexible batteries. Recently, advanced wearable electronics require thinner and smaller batteries, that is, high energy density which should be realized in a flexible form. Even though Si anodes hold a great potential regarding the usable energies, pulverized Si particles from cracking/fracture and consequent delamination from current collectors hinders further utilization of Si anodes in flexible batteries. This stress-resilient design of Si anodes could boot up the performances of flexible batteries, as demonstrated in this paper for sustainable structures even with delivering the high capacity.

How is the practicability of this method and its performance compared to the existing nano-engineered Si anode structures?

Response: As the reviewer pointed out, it is important to compare the practicability of our method and its performance with the existing technologies and nanostructured Si materials. In the newly added **Supplementary Note 1** in the revised manuscript, we described differentiate points in our preparation method. Unlike the reported strategies, our proposed method could provide a straightforward but controllable synthesis of 2DSi materials (**Supplementary Note 2**). Furthermore, the electrochemical performances we achieved with 2DSi@C electrodes are far beyond that of previous 2D Si-based anodes. To make it clear as possible, we compare those data in the newly added **Supplementary Table 1** along with other representative nano-engineered Si anodes, which are strictly sorted out. As analogized from the summary chart, typical Si materials have a trade-off in their initial Coulombic efficiency and cycling/rate performances, while our 2DSi@C electrodes showed well-rounded properties which is attributed to stress-releasing structure. Achieving >90% of ICE and cycling stability at moderate current density (~95%) looks quite promising enough to be comparable to other Si/C composite electrodes.

Supplementary Table 1. Comparison of electrochemical performances of 2DSi-based anodes and other nano-engineered Si anodes. Note: ^aReferences in the manuscript and ^bReferences in the Supplementary Information. The displayed capacities are based on the active materials only.

Electrodes	ICE(%)	Current density (A g ⁻¹)	Capacity (mAh g ⁻¹) (after X cycles) Retention (%)	Loading levels (mg cm ⁻²)	References
SiMP@Graphene	93.2	1.5	1,400 (500) 85	0.8	30 ^a
Pomegranate Si/C	82	0.7-1.0	1,160 (1,000) 97	0.2	19 ^a

Hollow Si	77	1.5	1,420 (700) 57	0.1	17 ^a
Si-C yolk-shell	60	3.0	1,500 (1,000) 74	1.0	36 ^a
Si-embedded graphite/carbon hybride	92	0.25	500 (100) 96	6.5	18 ^a
Si/C secondary particle	83.5	4.0	1,243 (150) 91	1.0	7 ^b
Ultrathin Si nanosheet	79.4	3.0	865 (700) 92.3	2.12	14 ^a
Carbon-coated Si nanosheets	47.7	0.4	1,575.5 (500) 92	1.5	8 ^b
2DSi@C	92.3	0.4	1,914 (200) 94.9	0.5-1.1	This work
		2.0	1,145 (500) 62.6		

Other strategies have also been reported to release mechanical energy or stress in Si-based anodes. Yu *et al.* (Adv. Energy Mater. 2012, 2, 68-73) used soft PDMS as a substrate for Si nano sheets. During lithiation, due to the constraint of the soft substrates, Si nano sheet ripples and therefore releases the mechanical energy inside the Si nano sheet. However, the PDMS here only plays a mechanical role, and in order for better energy releasing performance, the PDMS needs to be at least one order of magnitude thicker than Si nano sheet. In 2DSi@C we reported here, the carbon coating layer plays both mechanical (mechanical energy release) and electrochemical (contributing to the capacity and rate performance) roles, and the thickness of the coating layers is approximately 1/10 of the Si nanosheet. In addition, the PDMS-based design generates the ripples at the lithiation stage and the ripples remain during the delithiation process; whereas in our 2DSi@C ripples occur in the delithiation stage and disappear in the lithiation stage. Thus, the stress-relaxation mechanisms in these two designs are also different.

Though a numerical simulation is used to help explain the stress resilient characteristic, a deeper understanding can strengthen this paper even more. For instance, does the rippling behavior depend on the thickness of the 2D Si nano sheet? In what situations will this formation of the rippling structure break down?

Response: From fundamental mechanics, we know that rippling relaxes the in-plane compressive energy in the carbon coating layer, penalized by its bending energy. The bending energy scales $\Pi_B \sim E_c h^3 \kappa^2$, where E_c , h and κ are the Young's modulus, thickness, and curvature of the coating layer, respectively. The curvature of the coating layer scales $\kappa \sim A/\lambda^2$, where A is the amplitude and λ is the wavelength of the ripples⁶. Though the energy form of the substrate is quite complex and analytically unavailable (which prevents us from obtaining an analytical solution of A and λ in terms of other material properties), the

bending energy of the coating layer suggests that a thick coating layer suppresses/disfavors the rippling (small κ and A) while a thin coating favors rippling of large amplitude (large κ and A). Thus, as the coating thickness increases to a certain point, rippling would not occur ($A \rightarrow 0$). In this case, the in-plane compressive energy of the carbon coating cannot be relaxed. We have performed finite element analysis on the relationship between the coating thickness and the rippling amplitude, which agrees with our reasoning here, as shown in the **Supplementary Fig.21 and Note 3**.

Supplementary Figure 21. Coating thickness dependent rippling structure analysis. The rippling amplitude as a function of the coating thickness, obtained by direct numerical simulations. Here H is the thickness of the Si nanosheet.

Reviewer #3 (Remarks to the Author):

The large volume change of Si electrode upon electrochemical cycling is always a critical challenge for a stable Li-ion battery. In this paper, authors introduce a novel battery designing concept taking advantage of mechanical mismatch-driven rippling in carbon-coated Si sheets. The mechanism of the deformation patterns is demonstrated by in-situ TEM observation and chemomechanical simulation. Importantly, the ripple formation can release the cycling induced stress, and might help improve the electrochemical performance of Li-ion battery. However, there are some important issues that the authors have to address before this paper is accepted for publication in Nature Communications.

1. In this paper, 2D Si sheet is coated by an amorphous 5-10 nm thick carbon layer. The author found that this carbon coating can well restrict the huge expansion of 2D Si sheet upon

the lithiation and form a rippling upon delithiation. The authors have to also confirm that the carbon coating at this thickness would not self-break or delaminate when it is restricting the huge expansion or forming a rippling, using either in-situ or numerical methods.

Response: As the reviewer pointed out, typical amorphous carbon layers might not be mechanically stable to bear the large expansion of 2DSi. In order to confirm the stability of carbon layers upon lithiation, TEM analysis was carried out on the fully lithiated 2DSi@C sheets as shown in **Supplementary Fig. 16** of the revised manuscript. Interestingly, carbon layers rather expanded by measuring the spacing values, but no obvious fracture or break was found upon the full lithiation of 2DSi@C sheets. Carbon layers tightly cover the lithiated 2DSi sheets with dense and uniform SEI layers, which demonstrates a good structural integrity.

Supplementary Figure 16. *Ex situ* TEM analysis on fully lithiated 2DSi@C. (a) XRD pattern of fully lithiated 2DSi@C electrode. (b) Comparison of carbon interspacing values before and after lithiation. (c-f) TEM images of fully lithiated 2DSi@C sheet, which demonstrates no obvious fracture or break was not found on the carbon layers and, rather dense and uniform SEI layers were formed. The scale bars are 200 nm, 200 nm, 5 nm, and 10 nm for (c-f), respectively.

As discussed in 2) below, a thin coating layer may lead to the fracture of the coating layer during lithiation, while a thick coating layer may generate large tensile stress inside the nanosheet during delithiation, possibly causing pulverization. The in-plane stress generated in the coating layer and the Si nanosheet is due to the shear force at the interface, which may

cause delamination of the coating layer from the Si nanosheet. It turns out that the thickness of the coating layer in our experiments falls in a proper regime that does not cause either fracture or delamination.

2. The thickness of carbon coating is a critical factor to affect the ripple formation and stress release within the electrode. Is it possible to find out an optimized thickness of carbon coating by numerical simulation?

Response: Indeed, the thickness of carbon coating is a critical factor that affects the ripple formation and stress-releasing in the electrode. For the design purpose, one surely can optimize the thickness of the coating layer by numerical simulations.

During lithiation, the coating layer is subjected to tension while the Si in compression. As tension has potential to cause fracture, we focus our analysis on the coating layer. Assuming the lithiation induced that mismatched strain is ϵ_m , the mismatch generates a tensile stress in the coating layer that approximately scales $\sigma_c \sim \epsilon_m/h$ (see **Supplementary Fig. 22** below and **Note 3**). For very thin coating, σ_c could well exceed the fracture strength of the coating, causing cracking in the coating.

During delithiation, the coating layer is subjected to compression while the Si nano sheet in tension, so we focus our analysis on the Si nano sheet. The tensile stress of the nanosheet in the post buckling state of the coating is very complex and analytically unviable. However, one can easily see that the in-plane stress of the Si nano-sheet scales by $\sigma_I^{Si} \sim E_{Si}(\epsilon_0 - A^2/\lambda^2)$, where ϵ_0 is the chemical strain due to the delithiation inside the Si nanosheet, and A^2/λ^2 is the strain accommodated by the rippling morphology. The difference between the two is the actual tensile strain of the fully delithiated state. As discussed in Supplementary Fig. 21, thicker coating layer corresponds to smaller rippling amplitude (smaller A/λ), which causes very large in-plane tensile stress inside the Si nanosheet (see **Supplementary Fig. 22**).

The above scaling analysis shows that there exists an optimal thickness regime within which fracture of both the coating layer and the Si nanosheet can be avoided. Searching for the optimal value may be computational intensive, while this yields no further mechanistic understanding in the rippling phenomenon. We added several sentences to address this issue in the main text and leave the detailed mechanics analysis of the optimal thickness for future studies.

Supplementary Figure 22. In-plane stress analysis on 2DSi@C depending on the coating thickness. The first principal stress in the carbon coating layer (σ_c) during lithiation and that in the Si nanosheet (σ_{si}) during delithiation as a function of the coating thickness (h). Here H is the thickness of the Si nanosheet. Increasing the thickness of the coating layer reduces the tensile stress in the coating layer during lithiation but increases the tensile stress in the Si nanosheet during delithiation, which suggests the existence of an optimal thickness at which the tensile stresses in both the coating layer and the Si nanosheet are controlled in levels that avoid fracture.

3. In the Fig. 3(c), we can find the carbon coating will induce a field of tension stress within the 2DSi@C upon delithiation, while, for 2DSi sheet without carbon coating, it is basically stress-free state. It can be also confirmed by Fig. 4 (the 5th plots in 2nd and 4th rows). The question is: will this high tensile stress generate the cracks within the Si sheet? The authors should have a discussion about this tensile stress developed in 2DSi@C.

Response: In bare 2D Si nanosheet, lithiation generates tension on the surface (similar to other nanostructures like Si nanoparticles), as shown in figures below. On the contrary, in 2DSi@C nanosheet, during delithiation, the constraining effect of the coating layer generates tension in Si nanosheet, as pointed out by the reviewer. Since the tensile stress leads to crack formation and propagation and fracture eventually, the “dangerous” states are the lithiation stage in bare 2D Si and the delithiation stage in 2DSi@C. Comparing these two “dangerous” states, as shown in Figure R1, measured by the first principal stress (σ_1), the regions of high

tension reduced substantially in 2DSi@C, which explains the better electrochemical performance of 2DSi@C compared to bare 2DSi.

Figure R1. Comparison of stress state in (a) 2DSi during lithiation and in (b) 2DSi@C during delithiation.

4. In this paper, the simulation well explains the experimental observations. However, the details about the modelling are missing. I would recommend that one more section in “Method” discussing about the numerical modelling would make the simulation result more convincing.

Response: We appreciate this constructive question from the reviewer. The section has been added in “Method”.

Methods

Chemomechanical modeling. We employ a recently developed chemo-mechanical model to simulate the lithiation/delithiation process in bare Si nano sheet and 2DSi@C. For Si, in the finite-strain framework, lithiation induced deformation consists of the stretch rates and the spin rates. The total stretch rate is additive of the three components, the chemical ($\dot{\epsilon}_{ij}^c$), elastic ($\dot{\epsilon}_{ij}^e$), and plastic ($\dot{\epsilon}_{ij}^p$) one, $\dot{\epsilon}_{ij} = \dot{\epsilon}_{ij}^c + \dot{\epsilon}_{ij}^e + \dot{\epsilon}_{ij}^p$. The chemical stretch rate is assumed to be

proportional to the increment of the normalized lithium concentration, $\dot{\epsilon}_{ij}^c = \beta_{ij} \dot{c}$. The diagonal tensor, β_{ij} , represents the lithiation induced expansion. We set $\beta_{11} = \beta_{22} = \beta_{33} = 0.6$, and $\beta_{ij} = 0$ for the other entries. The elastic stretch rate, $\dot{\epsilon}_{ij}^e$, obeys Hooke’s law with the stiffness tensor, C_{ijkl} , depending on 2 independent material constants (i.e., Young’s modulus E and Poisson’s ratio ν). For the intermediate states of charge, the stiffness tensor is assumed to be linearly dependent on lithium concentration, interpolated by these two extreme states. The plastic stretch rate, $\dot{\epsilon}_{ij}^p$, obeys the classic $J2$ -flow rule. Namely, plastic yielding

occurs when the equivalent stress, $\sigma_{eq} = (3s_{ij} s_{ij}/2)^{1/2}$, reaches the yield strength. Here

$s_{ij} = \sigma_{ij} - \sigma_{kk} \delta_{ij}/3$ is the deviatoric part of Cauchy stress, σ_{ij} , and $\dot{\epsilon}_{ij}^p$ is proportional to s_{ij} . Note that we assume that both chemical and plastic deformations are spin-free. For carbon coating layer, we ignore the changes caused by lithiation/delithiation, and assume it deforms elastically.

In lithiation in crystalline Si (2DSi@C) and the first-step lithiation in amorphous Si (bare 2DSi), a critical feature is the formation of sharp interphase that separates the Li-poor and Li-rich domains. To capture this feature, lithium diffusivity is set to be nonlinearly dependent on lithium concentration: $D = D_0[1/(1 - c) - 2\alpha c]$, where D_0 and α are tunable constants to control the interphase profile between the Li-rich and Li-poor. The two-step lithiation in amorphous Si is realized by setting different Dirichlet boundary conditions (i.e. $c=0.67$ for $\text{Li}_{2.5}\text{Si}$ and $c=1$ for $\text{Li}_{3.75}\text{Si}$).

This chemo-mechanical model is numerically implemented in the finite element package ABAQUS/standard, with user subroutine UMATHT to realize the sharp interphase. For pure amorphous Si, we set $E_{aSi} = 100\text{GPa}$ and $\nu_{aSi} = 0.22$, while for pure crystalline Si $E_{cSi} = 160\text{GPa}$ and $\nu_{cSi} = 0.22$. For fully lithiated Si, we set $E_{Si}' = 25\text{GPa}$ and $\nu_{Si} = 0.22$. Yield strength of Si is set to be 1GPa . For carbon coating layer, we set $E_c = 250\text{GPa}$, $\nu_c = 0.25$ and $\sigma_{cY} = 30\text{GPa}$.

5. Since there is no discussion about the numerical modelling, I cannot see if the modelling of the lithiation/delithiation of the Si electrode is appropriate or not. Usually, the modellings of the lithiation/delithiation of the amorphous Si and polycrystalline Si electrodes should be different (2012, H. Yang, Nano letters, 12 (4), 1953 & 2016, R. Xu, Extreme Mechanics Letters, 8, 13-21 & 2015, C.V. Di Leo International Journal of Solids and Structures 67 283-296). Since the 2DSi is amorphous Si and 2DSi@C contains polycrystalline Si, has the difference between lithiations of the amorphous Si and polycrystalline Si been considered during the modelling?

Response: We thank to the reviewer's comment. Two-step lithiation is considered in the first lithiation of amorphous Si (bare 2DSi) in our simulations. As shown in Fig. 4a (2nd plot), there is a clear sharp interphase Li-rich phase ($c=0.67$, $\text{Li}_{2.5}\text{Si}$) and Li-poor phase (pure Si), while during the second step of lithiation diffuse interphase is employed. In our simulations, the two-step lithiation is realized by setting different Dirichlet boundary conditions (i.e. $c=0.67$ for $\text{Li}_{2.5}\text{Si}$ and $c=1$ for $\text{Li}_{3.75}\text{Si}$). Sharp interphase is also considered in lithiation of polycrystalline Si (2DSi@C) in our simulations. We have added the details of the simulations in Method.

6. In Page 5 Line 124: What does the "40Ag-1" correspond to the Crate? Since the authors used the Crate to describe the cyclic rate in the rest of the manuscript, it is better to also use the Crate here.

Response: As the reviewer pointed out, we changed the "40 A g-1" to "20 C-rate" in the revised manuscript. We unified all the units for current density into C-rate to make it clear as possible for potential readers. Also, we changed the x-axis in Fig. 1f into C-rate.

7. Are the specific capacities of 2DSi and 2DSi@C calculated by the weight of active materials? For the specific capacity of 2DSi@C, is the weight of carbon coating taken into account? The authors should clarify this.

Response: All the specific capacities described in this manuscript were calculated based on the weight of only 2DSi or 2DSi@C active materials and we did not exclude the weight of carbon coating layers (~7 wt%) when calculating the specific capacity of 2DSi@C. Capacity and weight contribution to entire electrode from carbon layers is almost negligible compared to Si anodes. When we take carbon contribution to capacity and weight into consideration, the specific capacity of 2DSi@C is still lower than that from theoretical calculation, which support our rationale in the original manuscript “The compressive stress may also cause lithiation retardation and possibly makes the inner regions of the 2DSi@C flake electrochemically inaccessible at the applied bias, which explains the slightly lower specific capacity of 2DSi@C compared to the bare 2DSi (Fig. 1d)^{42,43}.” This information was added in the Method section in the revised manuscript as follows:

All the specific capacities described above were calculated based on the weight of active materials. In particular, specific capacity of 2DSi@C were calculated by including the weight of carbon coating layers.

8. The authors claim the ripple formation also occurs in the liquid electrolyte system. What is the electrolyte the authors used in this paper? Were any additives added into the electrolyte? It is very important because the electrolyte would significantly change the electrochemical performance of the Si electrode. The authors should add more details and necessary information in the electrochemical characterization section to make the experiments more convincing.

Response: As the reviewer pointed out, electrochemical properties of LIBs strongly depend on the electrolyte composition (combination of linear and cyclic carbonate) and kind of additives. Therefore, we added the detailed information on electrolytes including compositions and additives used for electrochemical test along with other components for cell assembly in Method section in the revised manuscript as follows:

Electrochemical characterization. Electrochemical test was evaluated using 2032 coin-type cell. The working electrode was composed of 80 wt% as an active material, 10 wt% binder ((poly(acrylic acid)/sodium carboxymethyl cellulose, PAA/CMC, 50 wt%/50 wt%, Sigma-Aldrich)) and 10 wt% super-P (TIMCAL), which were prepared by slurry coating method. The loading levels of series of 2DSi was $0.5\text{-}1.1\text{ mg cm}^{-2}$. The coin cells, composed of 2D Si electrodes, polymer separator (Celgard 2400), and a lithium counter electrode were assembled in an argon-filled glove box. The electrolyte was a 1.3 M LiPF_6 solution in ethylene carbonate/diethyl carbonate (3/7 vol%) with 10 wt% fluorinated ethylene carbonate additives included to improve the cycling stability. All the specific capacities described above were calculated based on the weight of active materials. In particular, specific capacity of 2DSi@C were calculated by including the weight of carbon coating layers. For cathode, LiCoO_2 (LCO, L&F Inc.) powder was used as active material with its composition of 95:2:3 (LCO : polyvinylidene fluoride (PVdF) : super-P) and with a loading level of $\sim 20\text{ mg cm}^{-2}$. For full cell, theoretical N/P ratio was ~ 1.1 . The operating voltage window was from 0.005-1.5V at a 0.05-20 C-rate for anode half-cell, 3.0-4.3V for cathode half-cell, and 2.5-4.2V for full-cell. Electrochemical properties were measured using a cycle tester (WBCS3000 battery systems, Wonatech).

9. In the section from Line 229 to Line 244, the authors misused the “lithiation” and “charge”. For example, Line 234, “the Si was fully charged to the form of $\text{Li}_{15}\text{Si}_4$ ” should be “the Si was fully discharged to the form of $\text{Li}_{15}\text{Si}_4$ ”. Same misuses can be found elsewhere in this section. The authors have to modify this.

Response: We apologize for such misuses regarding the terms ‘lithiation/delithiation or discharge/charge’. As the reviewer already knows, most of fundamental studies are based on the Li/Si situation, that is, half-cell. Thus, lithium intake for Si anodes should be noted as discharge (lithiation) process, since Si anodes are used as cathode, while the lithium metal as

anode. We unified every related term to prevent possible confusion to potential readers in the revised manuscript. Also, we have noted this issue in the Page 3 and Line 19 as follows; ‘while its nanoscale thickness (out-of-plane) mitigates stress generation during lithiation (discharge).’

10. Please add references for the discussion in Line 234 “the resistance gradually decreased due to the formation of crystalline lithiated Si (Li_xSi) that enhances charge transfer kinetics”

Response: We added appropriate references to support above explanation. Please see the reference 47, 48 (newly added) in the revised manuscript. As a footnote, the net charge of the Si atoms becomes more negative with increased Li content of the Li-Si phases (typically in crystalline phase), compared with pure Si by Bader charge analysis⁴⁸. Importantly, Si anode undergoes a series of phase transformation to $\text{Li}_{3.75}\text{Si}$ (or $\text{Li}_{3.75+\delta}\text{Si}$), whereby two amorphous Li_xSi have minimum diffusion kinetics or charge transfer as showing ‘W’ type curves for diffusion coefficient¹.

1) Ding, N. *et al.* Determination of diffusion coefficient of lithium ions in nano-Si. *Solid State Ion.* **180**, 222-225 (2009).

Reviewers' comments:

Reviewer #3 (Remarks to the Author):

The authors provided the thoughtful responses to my comments. Most defects of this work have been solved. I believe now this manuscript could be accepted by the Nature Communications.

Reviewer #4 (Remarks to the Author):

The authors have coated 2D Si nanosheets with carbon to provide mechanical constraint during electrochemical cycling as a Li-ion battery electrode. These structures exhibit an interesting rippling behavior during cycling, which could represent a mechanism for releasing cycling induced stress.

Overall, the paper is well written, clear, and the results are interesting. However, prior to publication, it should address the following issues:

1. Is a 50 nm thick film really considered 2D Si? Will this have different performance than e.g., a 50 nm thick sputtered film? Similar thickness Si materials have been examined in literature (e.g., by sputtering) so the authors must comment on the novelty of these systems relative to existing ones from a fundamental perspective. Moreover, other studies have explicitly investigated the effects of inactive confining layers on lithiation of Si (e.g., core/shell structures). In light of these existing studies, the authors must clearly lay out the novelty of these studies that set them apart as a new paradigm relative to existing studies to warrant publication in this journal. How will these studies influence thinking in the field?

2. On page 3, the authors detail existing designs for mitigating stress in the literature and state "However, it is highly desired to develop stress-releasing materials in the first place without compromising other electrochemical properties." This statement is unclear. How are the previously mentioned approaches (e.g., porous structures) not "stress-releasing"?

3. The authors state that bare 2D Si flakes exhibit isotropic expansion. Is this really isotropic expansion in the bare 2D flakes? According to the measurements of the lateral expansion (31.9% as compared to the theoretical ~55%), it does not seem isotropic. Please comment. For instance, could there be inhomogeneous lithiation through the thickness or laterally?

4. The authors suggest that the carbon layers provide constraint during cycling, which mitigates tensile stresses in the 2D Si@C structures. This statement is true to a certain extent but may be somewhat misleading. From my intuition, even if the carbon layer constrains the deformation, I would still expect large tensile stresses during delithiation of the 2DSi@C due to plasticity during cycling and the associated large levels of deformation in these systems. It is even possible/likely that the maximum stresses will be similar in both systems due to the large strains and plastic deformation in the lithiated silicon (stresses are limited by yield strength). Indeed, Figure 4 seems to indicate this intuition. Namely, according to the scale bar, it appears the maximum levels of stresses during de-lithiation are around 2 GPa in both systems. Is this correct? Admittedly, it is somewhat difficult for me to view details of Figure 4 in the submitted version (e.g., colors). If so, this point needs to be address/clarified.

5. In the caption of Figure 1f, please clarify what is meant by rate capability (e.g., after how many cycles).

6. In the caption of Figure 1h, please clarify what the full cell system is (what is the cathode material).

7. Related to the simulations shown in Figure 2, the authors need to provide an explanation of how

the rippling originates in the simulation. For instance, is it possible rippling occurred due to natural perturbations in the simulations (e.g., due to finite mesh size). If so, they need to clarify if the mesh size artificially influenced their results. Similar, was a buckling analysis employed? Were some perturbations intentionally implemented?, etc.

Responses to all reviewer comments for Manuscript ID: NCOMMS-18-01149

*Blue: Words in the original paper, Red: Modified version.

Reviewer #3 (Remarks to the Author):

The authors provided the thoughtful responses to my comments. Most defects of this work have been solved. I believe now this manuscript could be accepted by the Nature Communications.

Response: We appreciate the reviewer's strongly positive comments.

Reviewer #4 (Remarks to the Author):

The authors have coated 2D Si nanosheets with carbon to provide mechanical constraint during electrochemical cycling as a Li-ion battery electrode. These structures exhibit an interesting rippling behavior during cycling, which could represent a mechanism for releasing cycling induced stress.

Overall, the paper is well written, clear, and the results are interesting. However, prior to publication, it should address the following issues:

1. **1)** Is a 50 nm thick film really considered 2D Si? **2)** Will this have different performance than e.g., a 50 nm thick sputtered film? **3)** Similar thickness Si materials have been examined in literature (e.g., by sputtering) so the authors must comment on the novelty of these systems relative to existing ones from a fundamental perspective. Moreover, other studies have explicitly investigated the effects of inactive confining layers on lithiation of Si (e.g., core/shell structures). In light of these existing studies, the authors must clearly lay out the novelty of these studies that set them apart as a new paradigm relative to existing studies to warrant publication in this journal. **4)** How will these studies influence thinking in the field?

Response: We appreciate the reviewer's helpful comments. Below is our point-to-point response to the reviewer's comments numbered above.

1) First of all, the prepared material in our study is not 50-nm-thick film, but large area 2D 50-nm-thick flakes or nanosheets according to the different classification principles as shown in Fig. 1. In general, dimensional grouping such as 0D, 1D, 2D and 3D materials depends on the dimension of their bulk counterparts. Since the lateral dimension of our material is over 10 μm while the thickness is approximately 50 nm, we believe it makes sense to group the material as 2D nano Si, given the large aspect ratio (~ 200). This dimensional grouping is consistent with carbon nanotubes. Typically, a carbon nanotube

is of 1 nm in diameter, but can be 1 μm . Accordingly, we categorize the carbon nanotube as 1D nanomaterial.

- 2) Yes, our 2D Si anodes have distinct performance from a 50-nm-thick thin film anode prepared by sputtering. The two systems differ in three aspects. First, the sputtered systems have an integrated electrode structure that needs no binder or conducting agents. The electrons are directly transferred to the Si thin film anodes through the current collector, which might enhance the electronic conductivity. However, the exhibited electrochemical performance of our 2D Si anodes stems from the composite electrode, consisting of binders and conductive carbon, a much complex system. Second, by extension, the Si thin film anodes prepared by sputtering only allow ionic diffusion through upper side and bottom contact area, whereas our 2D Si anodes allow ionic diffusion from all directions. Third, regardless of the outermost protection layers, Si thin film anodes always have a mechanical constraint upon alloying/dealloying reaction from physically contacted bottom surface to the current collector. Due to the limited areal capacity of sputtered Si thin films, it is not suitable for high energy density batteries. However, our 2D-Si showed tunable areal capacity for full-cell practical battery application. Thus, it is expected that they should surely exhibit very different electrochemical properties.
- 3) As we detailed previous studies on Si nanomaterials in a similar size in the introduction, various nanomaterials have distinct electrochemical performances over bulk Si anodes. 2D Si anodes facilitate large-area (in-plane) preparation to increase the material tap density to a certain extent compared to 0D (SiNPs) or 1D (SiNWs or SiNTs), while its nanoscale thickness (out-of-plane) still mitigates stress generation during lithiation and prevents fracture. This fact makes 2D Si much fascinating for a practical purpose. Besides, previously reported 2D Si anodes have much thinner 2D frames less than 10 nm without controllability on size and thickness (*ACS Nano* **2**, 2843-2851 (2016), *Adv. Mater.* **29**, 1701777 (2017) and *Small* **14**, 1703361, (2018)), but their electrochemical properties are still far below our material. This outcome may be largely attributed to the unique stress-releasing motion in ripple formation and systematically optimized thickness of 2D Si nanosheet for increased the material tap density compared to a few nanometer-thick Si nanosheets. Furthermore, we introduce a scalable synthesis protocol to prepare our 2D Si anodes via recyclable salt templating, which renders the lateral size of deposited Si also tunable. By synthesizing 10- μm -sized uniform NaCl salt template in a kilogram batch scale which can be recycled later, we applied Si CVD method in a rotary furnace to uniformly coat amorphous Si onto the template with a tunable thickness in a kilogram batch scale as well. In this way, we could say that our synthetic protocol and electrochemical properties/behaviors of the resulting 2D Si anodes are innovative compared to previous reports.

Further, as the focus of this manuscript, we took advantage of the high-volume change in Si anodes during electrochemical cycling rather than fighting against such a large deformation in most previous studies. Introducing a rigid coating layer or void spaces could address the critical issues in alloy-type anodes with high degree of lithiation. However, there has been few attempts to design the Si anodes in its deformed state. The

large deformation materials expand over than 300% in volume. When the surface constraint presents, especially for 2DSi materials, anisotropic expansions (limited expansion to lateral direction) occur. During delithiation of stress-free state, isotropic contraction occurs in all the directions, leading to the formation of rippled patterns on Si sheets, which effectively releases the stress and maintains the overall morphology for many cycles. This unique phenomenon arises because of the mechanical mismatch in Si and carbon layers as well as the 2D nature of the structure, which is absent in anode materials.

In contrast, in bare 2D Si nanosheet, lithiation generates tension on the surface (similarly as other nanostructures like Si nanoparticles), as shown in figures below. The tensile stress leads to surface crack nucleation and eventually pulverization. On the contrary, in our 2DSi@C nanosheet, the constraining effect of the carbon coating layer converts the tension in bare Si to compression on the surface during lithiation, which suppresses crack formation and propagation and therefore avoids pulverization. During delithiation in 2DSi@C, tensile stress is generated in the Si nanosheet, while compression in the coating layer. However, due to the formation of ripples, significant mechanical energy is released during delithiation in 2DSi@C, and therefore tension in Si nanosheet is controlled and does not cause fracture either.

As the figures shown below, measured by the first principal stress (σ_1), the regions of high tension reduced substantially, which, we believe, is the reason of the better electrochemical performance of 2DSi@C compared to bare 2DSi. In all, the unique attribute of the 2DSi@C is that the carbon coating layer alters the mechanical stress-generation pathway seen in bare Si (*i.e.*, converting tension to compression in lithiation), and releases the mechanical energy in each cycle, making it chemomechanically very durable and electrochemically superior.

Figure R1. Comparison of stress state in (a) bare 2DSi during lithiation and in (b) 2DSi@C during delithiation.

This innovative stress-releasing idea is broadly applicable in other high-volume-change materials (*i.e.*, Ge, Sn, Sb, P, etc.) for lithium or sodium ion batteries. In particular, owing to the much larger atomic size of sodium ions than lithium ions, high-energy-density anode materials such as phosphorus (P) for sodium ion batteries undergo an even larger volumetric change (p to 500%) during electrochemical cycling than those for lithium ion batteries. Fighting against such a large volume change is very challenging, if not hopeless. Our stress-releasing idea opens a new avenue to mitigate the large-deformation induced

battery material degradation, leading toward highly durable high-performance electrode materials.

Further, inherent to our 2D Si anode material is its mechanical flexibility, which can be exploited for high energy density flexible batteries. Recently, advanced wearable electronics has attracted much attention, which requires to be powered by thinner and smaller batteries, i.e., batteries with high energy density and high mechanical flexibility. Though Si anodes hold great potential for their high energy density, large-volume-change induced pulverization and delamination from current collectors hinder their utilization as flexible batteries. This stress-resilient design of Si anodes could meet the requirements of both high energy density and high flexibility, thus holding the promise of new batteries for wearable electronics.

2. On page 3, the authors detail existing designs for mitigating stress in the literature and state “However, it is highly desired to develop stress-releasing materials in the first place without compromising other electrochemical properties.” This statement is unclear. How are the previously mentioned approaches (e.g., porous structures) not "stress-releasing"?

Response: We appreciate the reviewer’s helpful comment on our confusing statement. As the reviewer pointed out, the nanoscale designs or porous structures in Si anode are also aiming at stress-releasing upon alloying/dealloying reactions. Our original intention from the above statement is that, while most nanostructured Si and porous structures can also achieve extended cycle life and fast charge/discharge rate, these materials typically have low initial Coulombic efficiency and low tap density (relative to the electrode density), with only a few exceptions such as pomegranate-like Si/C clusters or Si nanolayer-embedded graphite composite anode. That says, it is critically important for battery materials to have well-balanced properties that are superior in nearly all the aspect. We have modified the statement accordingly in the revised manuscript (Page 3, line 17);

Yet, with only a few exceptions, the aforementioned materials failed to attain a high initial Coulombic efficiency (ICE) as well as improved tap density while maintaining its stress-releasing action for robust battery operation.

3. The authors state that bare 2D Si flakes exhibit isotropic expansion. Is this really isotropic expansion in the bare 2D flakes? According to the measurements of the lateral expansion (31.9% as compared to the theoretical ~55%), it does not seem isotropic. Please comment. For instance, could there be inhomogeneous lithiation through the thickness or laterally?

Response: We appreciate the reviewer’s helpful comments. There are three possible reasons for having a different expansion of 31.9% in experimental compared to ~55% in theoretical analysis. First, the images in Fig. S12 of the expanded bare 2D Si flakes could be distorted since the samples might have a just point contact with tungsten counter electrode. In other words, the flakes might be slightly tilted to give out-of-range values compared to theoretical expectation. Secondly, there might be an experimental variation on the volume expansion of samples. For example, in the case of amorphous Si sphere (*Nano Lett.* **13**, 758-764 (2013)), the authors stated “Of 26 spheres that were measured, the total volume expansion varied from 101 to 332% with an average of 204%. This range of volume expansion is probably due to

different experimental conditions, such as the quality of electrical contact or variations in the thickness of the solid electrolyte oxide layer, which result in different degrees of lithiation in each sphere.” In fact, the volume expansion of bare 2DSi in the set of Fig. S18 was about ~50%. However, the 2DSi@C flakes always showed very low expansion in lateral direction and large expansion in vertical direction that have a clear anisotropic expansion behavior. Third, the position of lithium source may play a role on the deformation of lithiated 2DSi. As we examined in Fig. S17, when lithium source was overlapped on part of the top surface of 2DSi, the fully lithiated 2DSi was on longer planer but wrapped. Since TEM only measured the projection, this may also explain the less lateral expansion than the theoretical one. To avoid this misleading result, we modify the description on lateral expansion values as follows in revised manuscript (Page 6, line 20);

In contrast, bare 2DSi without carbon coating layers swelled into the lateral direction as much as 31.9% after the first lithiation, while the average lateral expansion in other control experiment was approximately 50%. The experimental variation may arise from different experimental conditions (Supplementary Fig. 12).

4. The authors suggest that the carbon layers provide constraint during cycling, which mitigates tensile stresses in the 2D Si@C structures. This statement is true to a certain extent but may be somewhat misleading. From my intuition, even if the carbon layer constrains the deformation, I would still expect large tensile stresses during delithiation of the 2DSi@C due to plasticity during cycling and the associated large levels of deformation in these systems. It is even possible/likely that the maximum stresses will be similar in both systems due to the large strains and plastic deformation in the lithiated silicon (stresses are limited by yield strength). Indeed, Figure 4 seems to indicate this intuition. Namely, according to the scale bar, it appears the maximum levels of stresses during de-lithiation are around 2 GPa in both systems. Is this correct? Admittedly, it is somewhat difficult for me to view details of Figure 4 in the submitted version (e.g., colors). If so, this point needs to be address/clarified.

Response:

We appreciate the reviewer’s constructive comments. As pointed out by the reviewer, in bare 2D Si nanosheet, lithiation generates tension on the surface, whereas, in 2DSi@C nanosheet, during delithiation, the constraining effect of the coating layer generates tension in Si nanosheet. Since the tensile stress leads to crack formation and propagation and fracture eventually, the “dangerous” states are the lithiation stage in bare 2D Si and the delithiation stage in 2DSi@C. Comparing these two “dangerous” states, as Fig. R1 shown below, measured by the first principal stress (σ_1), the regions of high tension reduced substantially in 2DSi@C, due to the stress-releasing by the carbon-coating buckling, which explains the better electrochemical performance of 2DSi@C compared to bare 2DSi.

To clarify this point more clearly, we modify Fig. 4 by picking more representative snapshots during the lithiation/delithiation process to replace the original ones (the second ones in Fig. 4 a and b), as shown below.

Figure R1. Comparison of stress state in (a) bare 2DSi during lithiation and in (b) 2DSi@C during delithiation.

Figure 4 Comparisons of the morphological evolution between 2DSi and 2DSi@C by the chemomechanical modeling. Snapshots of deformation morphologies predicted by the chemomechanical model, showing (a, c) lithium concentration and (b, d) the first principal stress of 2DSi and 2DSi@C, respectively.

5. In the caption of Figure 1f, please clarify what is meant by rate capability (e.g., after how many cycles).

Response: We appreciate the reviewer's helpful comments. In order to clarify the meaning of rate capability and how we tested the cells at different C-rates, we modify the figure 1f and its caption in below image. In fact, we switched the C-rate from 0.2 C to 20 C every 5 cycles.

Figure 1. Physical and electrochemical characterization of 2DSi-based anodes. (a) SEM images of 2DSi with high and low magnifications. (b) A TEM image of 2DSi@C, inset shows the typical SAED patterns of polycrystalline Si. (c) A high magnification TEM image, showing 5-10 nm amorphous carbon layers coated on the 2DSi. (d-h) The side-by-side comparison of half-cell electrochemical performance between 2DSi and 2DSi@C electrodes on initial galvanostatic voltage profiles (d, inset: surface area results of 2DSi, 2DSi@C, and SiNP), capacity retention at 0.2 C-rate (e), **rate capability at different C-rate from 0.2C to 20C for each 5 cycles** (f), long-term stability at 1 C-rate (g), and **capacity retention of full-cell paired with LiCoO₂ cathode (h)**, respectively. Scale bars, 100 nm and 5 μ m (a); 500 nm (Inset: 2 1/nm) (b); and 5 nm (c).

6. In the caption of Figure 1h, please clarify what the full cell system is (what is the cathode material).

Response: We appreciate the reviewer's helpful comments. We modify the caption as follows; **capacity retention of full-cell paired with LiCoO₂ cathode (h)**.

7. Related to the simulations shown in Figure 2, the authors need to provide an explanation of how the rippling originates in the simulation. For instance, is it possible rippling occurred due to natural perturbations in the simulations (e.g., due to finite mesh size). If so, they need to

clarify if the mesh size artificially influenced their results. Similar, was a buckling analysis employed? Were some perturbations intentionally implemented?, etc.

Response:

In our simulations, quadratic elements were employed. As we examined, the influence of the mesh size on the buckling (e.g. amplitude and wave length) is negligible. There was not any intentional perturbation in our simulations. However, since we simulated the dynamic process of lithiation/delithiation, the delithiation from the delithiation source to the clamped end may played a role to trigger the buckling, as shown in Fig. S15. We also believe that the dynamic nature of our simulations may automatically introduce perturbation to the simulation system to induce rippling.

From fundamental mechanics, we understand that rippling relaxes the in-plane compressive energy in the carbon coating layer, penalized by its bending energy. The bending energy scales $\Pi_B \sim E_c h^3 \kappa^2$, where E_c , h and κ are the Young's modulus, thickness, and curvature of the coating layer. The curvature of the coating layer scales $\kappa \sim A/\lambda^2$, where A is the amplitude and λ is the wavelength of the ripples (Efimenko K, et al. Nature Materials 2005, 4(4), 293). Though the energy form of the substrate is quite complex and analytically unavailable (which presents us from obtaining an analytical solution of A and λ in terms of other material properties), the bending energy of the coating layer suggests that a thick coating layer suppresses/disfavors the rippling (small κ and A) while a thin coating favors rippling of large amplitude (large κ and A). Thus, as the coating thickness increases to a certain point, rippling would not occur ($A \rightarrow 0$). In this case, the in-plane compressive energy of the carbon coating cannot be relaxed. We have performed finite element analysis on the relationship between the coating thickness and the rippling amplitude, which agrees with our reasoning here, as shown in the figure below.

Figure R2. The rippling amplitude as a function of the coating thickness, obtained by direct numerical simulations. Here H is the thickness of the Si nanosheet.

REVIEWERS' COMMENTS:

Reviewer #4 (Remarks to the Author):

The authors have adequately addressed my concerns so the paper is now acceptable in my opinion.